# Amino acids catalyse RNA formation under ambient alkaline conditions

Saroj K. Rout [1,5], Sreekar Wunnava [1], Miroslav Krepl[2], Giuseppe Cassone [3], Judit E. Šponer [2], Christof B. Mast [1], Matthew W. Powner [4] & Dieter Braun [1] ✉

RNA and proteins are the foundation of life and a natural starting point to explore its origins. However, the prebiotic relationship between the two is asymmetric. While RNA evolved to assemble proteins from amino acids, a significant mirror-symmetric effect of amino acids to trigger the synthesis of RNA was missing. We describe ambient alkaline conditions where amino acids, without additional chemical activators, promote RNA copolymerisation more than 100-fold, starting from prebiotically plausible ribonucleoside-2′,3′-cyclic phosphates. The observed effect is explained by acid-base catalysis, with optimal efficiency at pH values near the amine $pK_{aH}$. The fold-change in oligomerisation yield is nucleobase-selective, resulting in increased compositional diversity necessary for subsequent molecular evolution and favouring the formation of natural 3′−5′ linkages. The elevated pH offers recycling of oligonucleotide sequences back to 2′,3′-cyclic phosphates, providing conditions for high-fidelity replication by templated ligation. The findings reveal a clear functional role of amino acids in the evolution of RNA earlier than previously assumed.

Life utilises an interconnected network of polymers—nucleic acids and proteins—to encode and replicate information as the basis for Darwinian evolution. Back extrapolation of evolution led to the most widely accepted hypothesis for abiogenesis[1–8], the RNA world hypothesis, in which life originated with RNA but no proteins[9–12]. This model appeared to be validated by the discovery of ribozymes[13–17], the ribosome, and RNA-catalysed RNA replication[18–20]. Whilst non-enzymatic processes to synthesise nucleotides and nucleic acids have been reported[6,11,21–33], the required length of catalytic RNAs poses a large barrier to the RNA world, and the chemical synthesis of RNA long enough to initiate self-replication cycles remains an open question. As a result, elucidating chemistry that can produce RNA capable of self-replication remains, despite decades of work, an essential goal for Origin of Life research.

RNA replication can occur via two primary mechanisms: base-by-base templated polymerisation akin to that carried out by protein polymerases[24,29] or templated ligation of short oligomers[31,32]. Current methods for templated polymerisation require high $Mg^{2+}$ concentrations, which in turn prevent subsequent strand separation, thus inhibiting downstream replication cycles. Recent work has demonstrated the possibility of ligating oligomers once they are long enough for hybridisation with lengths longer than four bases in the presence of low $Mg^{2+}$ concentration and from 2′,3′-cyclic phosphate activation, suggesting a potential pathway for ligation chain reaction cycles[33]. This will require a significant concentration and diversity of long enough oligomers to be available by non-templated polymerisation, which is the aim of this study. It should also be noted that highly efficient polymerisation should be avoided since long random sequences

[1]Systems Biophysics and Center for NanoScience, Faculty of Physics, Ludwig-Maximilians-Universität München, Geschwister-Scholl-Platz 1, 80539 Munich, Germany. [2]Institute of Biophysics of the Czech Academy of Sciences, Královopolská 135, 61200 Brno, Czech Republic. [3]Institute for Chemical-Physical Processes, National Research Council of Italy (IPCF-CNR), 98158 Messina, Italy. [4]Department of Chemistry, University College London, London WC1H 0AJ, UK. [5]Present address: Institute of Molecular Physical Science, ETH Zurich, 8093 Zurich, Switzerland. ✉e-mail: dieter.braun@lmu.de

would compete with replicated sequences, hindering information transfer and storage.

Biologically, the chemistry of nucleic acids is intimately associated with peptides and amino acids, but this vital connection is lost in the 'RNA-only world' model, and we suspected that the problems associated with the independent synthesis of RNA would be ameliorated by functional interactions with amino acids and peptides. Amino acids are a simple, plausible component of the early Earth's chemical inventory; therefore, the mutualistic interaction of amino acids and RNA could predate life. Indeed, amino acids and nucleotides can be accessed via the same divergent prebiotic chemistries[34,35]. However, RNA chemistry has not yet been successfully united with amino acids, and the prebiotic functional relationship between these molecules remains limited. While recent studies have explored the possibility of using amino acids as leaving groups in amino acid phosphoramidate nucleotides for RNA replication[36] and ribozyme assembly[37], these hybrid molecules rely on the prebiotically improbable activating agent EDC.

Ribonucleoside 2′,3′-cyclic phosphates (cNMP; $N$ = G,C,A,U) are the products of prebiotic synthesis[11,38,39], as well as the natural product of RNA cleavage[40]. We therefore considered the relationship between cNMPs and prebiotically plausible amino acids. The effect of amino acids on the polymerisation of only cAMP has previously been investigated by Orgel[6,25]. In these studies, various amine compounds, including amino acids, were examined under alkaline pH conditions. No variations of pH and no kinetic analyses were carried out, and therefore, reaction mechanisms could only be suggested. At room temperature, only glycine was tested, but no control without glycine was measured at the same pH, leaving the role of glycine undetermined. The only experiment with a specific control used a twofold excess of lysine in a fourfold excess of imidazole buffer at 85 °C. However, the constant addition of imidazole likely interfered with the reaction in the dry state, yielding only a twofold increase in polymerisation. Orgel et al. write[6], "Of the amino acids and peptides investigated as potential catalysts, only lysine and its oligomers proved effective, and only on heating at 85 °C". In contrast, we show here that without buffer molecules and at room temperature, amino acids increase polymerisation by more than hundered-fold.

Previously, significant RNA polymerisation from cNMPs was reported under alkaline conditions without the interference of added buffers[41], a scenario not investigated by Orgel. However, only short (≤4 nt) and G-rich oligomers were created. This G bias inhibits base pairing and blocks subsequent replication by templated ligation[12,42]. However, we suspected that the observed enhanced reactivity of G was due to its alkaline $pK_a$, which allowed nucleobase deprotonation to facilitate oligomerisation. Intrigued by the similarity in basicity ($pK_{aH}$) of amino acids, we speculated that amino acids would promote RNA oligomerisation as an intermolecular acid-base catalyst. This, we suspected, would ameliorate the natural sequence bias of (alkaline) cNMP polymerisation and therefore provide a direct link between the informational polymer RNA and amino acids.

To test our hypothesis, we investigated the role and effect of amino acids as prebiotic catalysts for the spontaneous yet selective polymerisation of cNMPs to yield (unbiased) sequences of RNA. We found that amino acids catalysed cNMP oligomerisation and, for the first time, efficient RNA polymerisation was observed at room temperature. Importantly, amino acid catalysis generated a sequence composition diversity that could not be achieved in their absence. This connection closes a missing link of cross-catalysis between both nucleic acids and amino acids and an interaction that underpins life (Fig. 1a, red).

## Results and Discussion
### The catalytic effect of amino acids on RNA oligomerisation
In the first set of experiments, a solution of cNMP (10 mM) and amino acid (0–5 equiv.) at pH 10 was rapidly (<30 min) dried under nitrogen flow on a glass slide and then incubated for 20 h at room temperature. A range of proteinogenic amino acids with both polar and nonpolar side chains were examined in the assay (Fig. S1). The oligomeric RNA products were extracted in nuclease-free water and quantified by calibrated HPLC-ESI-TOF mass spectrometry (Figs. S2 and 3). We observed a significant increase in both the oligomer length (Figs. 1b, S4–6 and Supplementary Data 1) and yield with the addition of amino acids (Fig. 1c).

Without amino acids, only cGMP (26%) furnished a significant yield of oligomers; the other nucleotides cUMP (0.7%), cAMP (0.17%) and cCMP (0.06%) gave only trace yields under the same conditions. The observed base imbalance and the short length of these oligomers make RNA hybridisation and therefore, templated replication highly unlikely. The addition of amino acids improved the reactivity of all four nucleotides, resulting in both increased and, importantly, more comparable yields. For example, adding valine (5 equiv.) resulted in 39%, 4.8%, 8.4% and 7.3% oligomers for G, U, A and C, respectively. We found that the increase in yield was largest for the least reactive nucleotides (i.e., C > A > U > G), and 122-, 49-, 6.8- and 1.5-fold increases in yield for C, A, U and G, respectively, were observed with valine (5 equiv.). It is also worth noting that we reported a 3.5% yield for cGMP polymerisation at 40 °C previously[41] but no polymerisation at 25 °C due to a too slow drying process in that study. In our current experiments, the yield increased to 26% without amino acids and to 39% with valine. Most importantly, however, a significantly enhanced yield was now observed for the other bases A, U and C.

The maximum enhancement was observed with aliphatic hydrophobic amino acids (Val, Leu and Ile), and we detected A- and C-homo-oligomers up to seven nucleotides long, whereas in the absence of these amino acids, only trace amounts of tri- or tetramers were detected (Fig. 1b). Also, the RNA pool contains a significant portion of sequences longer than 4-nucleotides that are capable of hybridising to a template[33]. For example, oligomers containing 5–7 nucleotides formed during cCMP polymerisation in the presence of hydrophobic amino acids contribute around 20% of the total yield. Interestingly, although Phe is hydrophobic, its aromatic side chain was not observed to significantly enhance the oligomerisation more than other polar amino acids such as Gly/Lys/His/Asp/Asn, in contrast to Val/Leu/Ile. Polar amino acids moderately enhanced the oligomerisation of cCMP, cAMP, and cUMP, but inhibited the polymerisation of cGMP. For example, the relative yields of RNA in the presence of Lys were 24-fold for cCMP, 22-fold for cAMP, 2-fold for cUMP, but 0.4-fold for cGMP. Various non-covalent interactions between nucleotides and specific amino acids may significantly interfere with the self-catalysis of the nucleobases by hindering the reactive molecular arrangements, as observed in cGMP polymerisation. These interactions may include stacking of Phe and interactions of Arg with the Hoogsteen face of G. We found an almost nucleotide-independent correlation between the aliphatic amino acid side chain hydrophobicity and their catalytic effect on RNA oligomerisation, with Leu/Ile > Val > Ala > Gly ≈ Lys > Asp (His/Arg/Asn/Pro/Phe). It appears that the physicochemical properties of the amino acid side chains influenced the yield and length distribution of the RNA products (Figs. S4 and 5). The heterogeneity in the morphology of the dried nucleotide-amino acid mixture did not significantly impact the oligomerisation (Fig. S6). However, hydrophobic amino acids could reduce the overall water content in the dry phase while also interacting with the nucleobases to facilitate the positioning of the amino acid at the cyclic phosphate.

[31]P-NMR analysis[41,43] of the RNA-oligomers revealed a higher prevalence of canonical 3′-5′ phosphodiester linkages compared to the non-natural 2′−5′ linkages in the presence of amino acids. In the presence of valine, oligo-A and oligo-C comprised 58%, and oligoU exhibited 66% of natural linkages (Fig. S7). The ratio of linkages in oligo-G was not quantified due to signal overlap and oligo-G aggregation.

## Factors affecting the efficiency of RNA oligomerisation

Due to the high hydrophilicity of ribonucleoside-2',3'-cyclic phosphates, the reaction mixture dried in air or nitrogen retains a considerable amount of water. We observed a significant weight loss of the nucleotides upon drying. The microscopic structure of the dry state suggests a flexible glassy state with lower chemical activity of water, enabling polymerisation. In an aqueous solution, no polymerisation was found.

To gain an understanding of the mechanism, we probed the pH dependence of the oligomerisation. Based on the above results, we chose to compare hydrophobic valine, charged lysine and glycine. Solutions of each cNMP (10 mM) with and without each amino acid (50 mM) were prepared across a broad pH range (pH 3–12) (Figs. 2a, S8–11 and Supplementary Data 2). The solutions were dried and incubated at room temperature for 20 h (Figs. S12–16).

The oligomerisation of cCMP and cAMP without amino acids was inefficient across this pH range, while the yield of cGMP (24%) and cUMP (1%) peaked around pH 10. This correlates with the fact that the basic-sites of G and U have a p$K_a$ around 9.3, while the basic-sites of the low-yielding cNMPs, C and A, have no p$K_a$ in the range investigated. While glycine and lysine most effectively enhance the oligomerisation yields for A, U and C around pH 9, 10, we found a reduction in polymerisation for G. In the presence of valine, the yields of all cNMP oligomerisation improved substantially and peaked around pH 9, 10. However, hydrophobic amino acids such as valine, leucine and isoleucine still enhanced the polymerisation for G at pH 10 (Fig. 1c). The fold-increase in yield is largest for A and C, likely because they lack an alkaline p$K_a$ and fail to polymerise by themselves.

We next performed a kinetic analysis of cCMP (10 mM) oligomerisation in the absence and presence of valine (50 mM) at initial pH 8–12 over 4 h in the dry state (Figs. 2b, S17 and 18). The initial rate of valine-catalysed RNA-oligomerisation peaked at pH 9, 10, close to the amine p$K_{aH}$ (9.6) of valine. At this pH, the relative ratio of the base (amine) and acid (ammonium) form of amino acid molecules is 1:1 in the reaction mixture. This leads to the maximum reaction rate because it enables a proton abstraction from the 5'–OH of the nucleophile and the concomitant protonation of the O2' or O3' oxygens of the substrate, which serve as the leaving group in the transphosphorylation reaction (Fig. 2d, e). These results suggest that amino acids catalyse RNA formation by acid-base catalysis, as described in Fig. 2d. Measurements of the pH over the drying process revealed that the added amino acid focuses the pH to its p$K_{aH}$, effectively making the initial pH distributions in Fig. 2a and b wider than in the dry state (Fig. S19). The maximal yield was observed when the concentration of amino acids was twice the nucleotide concentration (10 mM), and the concentration dependence was observed to saturate above 20 mM amino acid (Fig. 2c); it is likely that excess valine is sequestered through self-crystallisation in the dry state (Figs. S12–16). It should be noted that the polymerisation efficiency rises without amino acids at a high pH of 11 to 12, likely due to 5'–OH activation (Fig. 2a). However, this is ultimately limited by nucleotide and RNA degradation. At lower amino acid concentrations, the pH dependence persists (Fig. 2c). At higher concentrations, amino acids induce hydrolysis of 2',3'-cyclic phosphates during drying, suggesting an amino acid-dependent hydrolysis pathway already in the drying process (Fig. S18), reducing the overall yield. We normalised against these degradation effects in the analysis of

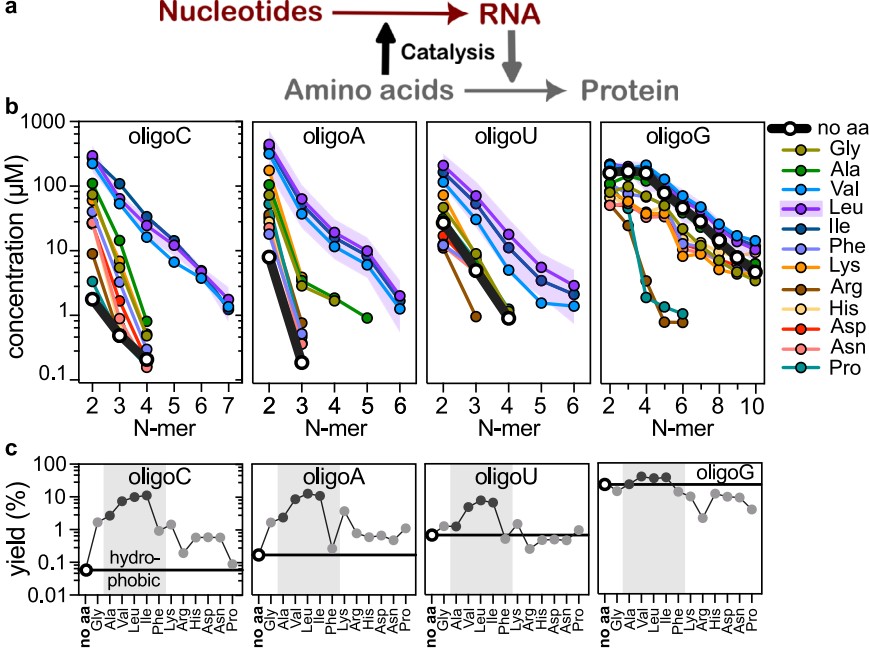

**Fig. 1 | Dry state oligomerisation of nucleoside 2',3'-cyclic phosphates catalysed by amino acids. a** RNA catalyses the formation of proteins from amino acids in the ribosome (grey). Our findings connect both molecule classes in an opposite catalytic direction at an earlier stage: amino acids catalyse the formation of RNA (red). **b** Concentrations of the oligomers of different lengths are plotted on a log scale for cCMP, cAMP, cUMP, and cGMP oligomerisation. With amino acids, the length-dependent concentrations are enhanced at least 100-fold, especially for bases A and C and around 10-fold for U. cGMP yields long oligomers already without amino acids. Controls without amino acids are shown in black (no aa), and amino acid-promoted reactions (corresponding three-lettered codes) are shown in other colours. Representative error bands (S.D.) are shown; for detailed errors, see Fig. S4. **c** The %-yields of the oligomers. The yields of RNA oligomerisation for different amino acids and nucleotides are calculated as the sum of nucleotides in oligomer form, i.e. the concentration of oligomers times their respective length. The shaded region highlights hydrophobic amino acids, with aliphatic chains shown in black, and all other amino acids are shown in grey. The data for no aa controls is visualised as a horizontal black line, indicating the many-fold increase of the overall yield. The effect of amino acid mixtures on RNA oligomerisation is presented in Fig. S5. All reactions were performed at 25 °C with 10 mM cNMP and 50 mM amino acid in 10 μl, at pH 10 for 20 h, and yields were quantified by reverse phase HPLC ESI-TOF mass spectrometry. Calibration against standards and fitting the isotope patterns were provided by a custom-written LabVIEW programme[58], detailed in Supplementary Data 1. Source data are provided for panels (**b**) and (**c**).

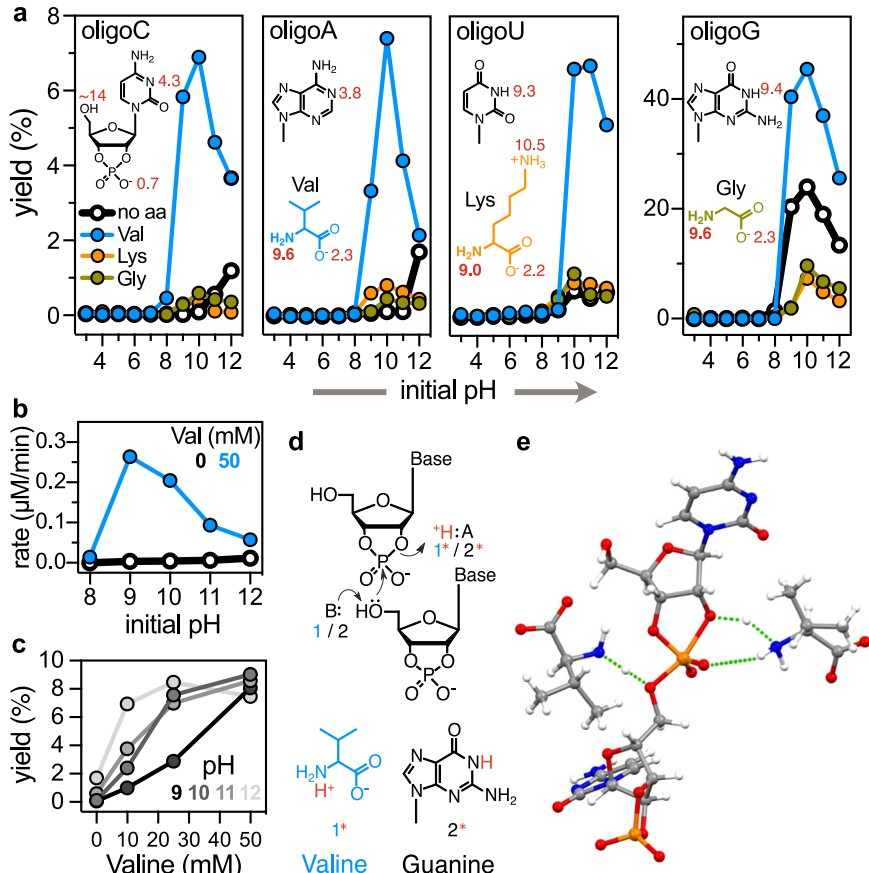

**Fig. 2 | Determinants of an efficient amino acid-catalysed oligomerisation of nucleoside-2′,3′-cyclic phosphates. a** pH dependence of the cNMP (10 mM) oligomerisation without amino acid (black) and with valine (blue), glycine (olive) and lysine (orange) (50 mM) are presented for the pH range 3–12. The chemical structures of nucleotides and amino acids are given as insets with p$K_a$ or p$K_{aH}$ values (red) of the (de)protonatable groups[60]. **b** The rate of cCMP (10 mM) oligomerisation in the absence and presence of valine (50 mM) is plotted. **c** Valine concentration-dependence (0–50 mM) of the cCMP (10 mM) oligomerisation is plotted at pH 9–12 to get insights into the mechanism of the amino acid-catalysed RNA oligomerisation. **d** Proposed mechanism of amino acid-catalysed RNA oligomerisation. In the dry state, due to the proximity and orientation effects, the amine group of the amino acid (1) may act as a general base to facilitate proton transfer

from the 5′–OH of the nucleotide, which attacks the cyclic phosphate of a second nucleotide whilst the ammonium moiety of zwitterionic amino acid (1*, protonated form of 1), may act as a general acid to facilitate proton transfer to the 2′–OH or 3′–OH. Acid-base catalysis can also be achieved by nucleobases, such as guanine (2 and its protonated form 2*) or uracil, as they both have p$K_a$ values close to the amine p$K_{aH}$ of the amino acids. The mechanism in the opposite direction can be followed for acid-base-catalysed RNA fragmentation[13]. **e** Proposed transition state complex of the reaction of valine-catalysed cCMP oligomerisation based on quantum chemical calculations using a simplified model (Fig. S20); colour codes: C −grey, N−blue, O−red, H−white, P−orange; H-bonds are shown in green dotted lines. All the experiments were performed at 25 °C. Isotope fits for the molecules in (**a**–**c**) are given in Supplementary Data 2. Source data are available for panels (**a**–**c**).

Fig. 2b since the kinetics is measured only after drying. Therefore, we interpret the peak at pH 9 to be the direct result of acid-base catalysis.

The same type of acid-base catalysis would support the self-oligomerisation of cGMP and cUMP, without amino acids, that also peak at the p$K_a$ of U and G (Fig. 2a, black lines). Moreover, it is noteworthy that the reaction in the reverse direction, namely the cleavage of RNA to 2′,3′-cyclic phosphates by a ribozyme[13,14], has been found to be general base catalysed via microscopically p$K_a$-shifted cytosine base at physiological pH.

### Amino acid selectivity of the RNA oligomerisation

Hydrophobic amino acids like Val, Leu, and Ile significantly enhanced RNA oligomerisation yields, whereas polar amino acids showed a moderate impact, as depicted in Fig. 1c. The extent of catalytic enhancement appeared to correlate with the hydrophobicity of the amino acid side chains. While the amino ends of the amino acids drive acid-base catalysis, the hydrophobic side chains may contribute to the spatial arrangement and potentially reduce the hydration around the 2′,3′-cyclic phosphates. The hydrophobic/hydrophilic characters are often linked to the dielectric properties

of a chemical environment[44]. To investigate whether these properties might directly affect the experimentally observed transphosphorylation reaction rates through the activation free energies, we conducted the following computations.

Using quantum chemical calculations and a simplified model of the reaction complex, we described the amino acid-catalysed attack of a primary alcohol on the phosphorus of a 2′, 3′ intramolecular phosphodiester linkage directly connected to a ribose moiety. This scenario represents a general case of an amino acid-catalysed reaction involving the 5′ OH group of a 2′, 3′ cyclic nucleotide and the cyclic phosphate of another nucleotide. As detailed in the Supplementary Information, our calculations revealed that the computed activation energies were only minimally affected by the dielectric constant of the medium, which was used to account for the bulk hydrophobicity or hydrophilicity of the chemical environment (Fig. S20). Thus, the differential enhancement of polymerisation is unlikely due to the local hydrophobicity provided by the amino acid side chain in the vicinity of the reaction centre. Therefore, we suspected that the different catalytic effect of amino acids arises from their preferential localisation around the catalytic sites of nucleotides.

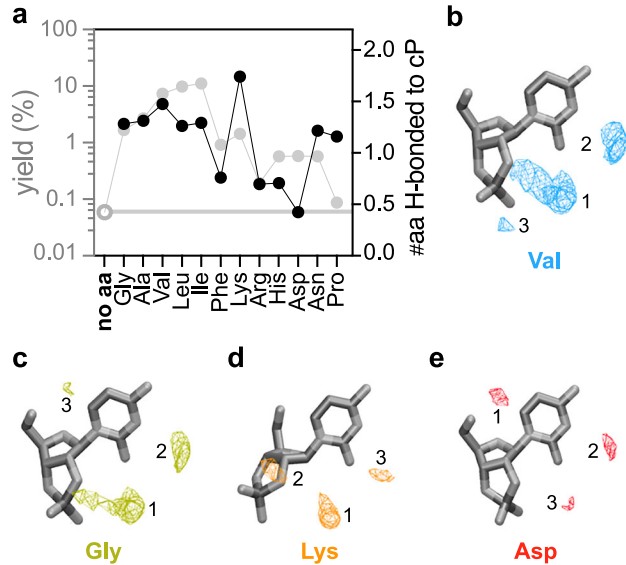

**Fig. 3 | Theoretical modelling to explain the experimentally observed amino acid selectivity in RNA oligomerisation. a** The yields of amino acid-catalysed 2′,3′-cCMP oligomerisation (grey) were correlated to the number of amino acids (black) interacting through hydrogen bonds with the cyclic phosphate group of the substrate nucleotide within an N…O distance of 3.5 Å (#aa H-bonded to cP) determined by classical molecular dynamics simulations. For further details, see Table S1. **b–e** The spatial distribution of amino acids around the 2′,3′-cCMP nucleotides in molecular dynamics simulations is represented by the three highest density regions (1–3), computed for valine (blue), glycine (olive), lysine (orange), and aspartate (red). For other amino acids, see Fig. S21. Source data for panel (**a**) is available. MD simulation trajectories are provided in ref. 59.

We then used classical molecular dynamics to assess the propensity of the amino group of the amino acids to form hydrogen bonds with the 2′,3′-cyclic phosphate for acid-base catalysis (Figs. 2d, e, 3, S21 and 22; Table S1). The simulations show that the amino groups of amino acids exhibiting higher catalytic activity in experiments (Val, Leu, Ile, Lys) have a noticeably higher preference to form hydrogen bonds with the phosphate group as well as with the RNA nucleobase. Quantum molecular dynamics simulations (Fig. S22) further explain the subtle difference observed between the classical molecular dynamics simulations and the experimental data for the case of Lys and Asn. These are the first attempts to theoretically understand the role of amino acids to enhance RNA polymerisation, and more detailed calculations are desirable but computationally expensive.

### Enhanced compositional diversity in amino acid-assisted G/C/A/U oligomerisation

An equalised distribution of bases is required for the replication of RNA, especially for templated ligation[33,42], which requires RNA sequences of at least four nucleotides to facilitate proper base pairing. A genetic copying process of this nature could enable the formation of longer RNA and increase the likelihood of discovering functional sequences of length comparable to tRNA molecules, thereby supporting the concept of Darwinian molecular evolution. However, the co-polymerisation of cNMPs observed so far yielded compositionally restricted oligomers, which would prevent base pairing to yield double-stranded RNA. This problem was ameliorated by the catalysis of amino acids. We found that amino acids promoted a compositionally more diverse copolymerisation of cG/CMP, cA/UMP and cG/C/A/UMP nucleotides upon drying cNMPs mixtures (total concentration 40 mM) with amino acids (Val or Leu; 2.5 equiv.) for 20 h.

The cA/UMP polymerisation without valine at 25 °C generated a flat concentration distribution, indicating a prevalence of

homopolymers, while the addition of valine produced a pronounced peak structure (Figs. 4a and S23–25), characteristic of random oligomerisation with a balanced composition of the bases. For example, the 3mer A₂U is composed of AAU, AUA, UAA, leading to a threefold enhanced measured concentration in mass spectrometry relative to AAA or UUU. A fully random sequence distribution is shown by the blue solid lines in Fig. 4. Similarly, cG/CMP-polymerisation at 25 °C produced a biased pool dominated by G-rich oligomers without amino acids, but the addition of valine enhanced the copolymerisation of cCMP and moved the composition closer to a random distribution of the bases (Figs. S23, S26–29). It is noteworthy that valine increased the yields of several oligomers, like AAUU and GGCC, by more than 20-fold. At 4 °C, the copolymerisation of cA/UMP (Figs. S24, 25: b) and cG/CMP (Figs. S26–29: b) exhibited similar effects to those observed at ambient temperature, though the yields were relatively lower due to slower reaction kinetics. However, the reduced hydrolysis at this temperature led to preserving more 2′,3′-cyclic phosphates. Additionally, the copolymerisation of cA/UMP (Fig. 4a) and cG/CMP (Figs. S27 and S29) resulted in similar concentrations of 6-mers and 3-mers, which contradicts Fig. 1b. This observation may indicate an aspect of efficient copolymerisation, where nucleobases (G and U), like amino acids, can catalyse copolymerisation. This effect is more noticeable for 6-mers, as they exhibit more variants than 3-mers.

The copolymerisation from the full base set G/C/A/U (with Val, Gly or Lys) confirmed this trend. Here, we even found that for 2–5-mers, the fraction of G was reduced (Fig. 4b). For example, the relative yield of GGGG decreased by a factor of three with valine catalysis, but the yield of GGHH and GHHH increased by a factor of four and eight, respectively, where H = {A, C, U}. The compositions of all possible oligomers and their relative yield differences are shown in Figs. 4c–e and S30–32. These demonstrate a polymerisation enhancement following C > A > U > G by amino acid catalysis (Fig. 1c), inverting the intrinsic yield trend observed without amino acid catalysis. The reduced incorporation of G due to self-catalysis, along with the similar absolute yields for other nucleotides in the presence of valine, decreases the G-bias in the oligomer pool, thereby enhancing sequence diversity.

Oligomers with 2′,3′-cyclic phosphates termini confer a functional advantage for the replicative evolutionary cycles because they remain activated to trigger templated ligation[33]. An efficient replication requires an RNA pool of high yield and compositional diversity. Additionally, the fidelity of templated replication peaks for sequences shorter than 8-mers[33]. Our results demonstrated that such a pool can be generated by the amino acid-catalysed RNA oligomerisation—for example, the diverse G/C oligomer pool with valine contains about 33% of the total oligomers with 2′,3′-cyclic phosphate termini.

Furthermore, the high selectivity of templated ligation can help in purifying double-stranded 3′-5′ linkages when combined with the recycling of rapidly hydrolysing 2′-5′ linked single-stranded RNA (Fig. S33)[45]. We probed alkaline hydrolysis of G/C oligomers at pH 10–13 and 5–25 °C and found that the oligomers remain activated with 2′,3′-cyclic phosphates even after 24 h (Figs. S34–38). Importantly, RNA fragmentation was observed to yield more cGMP and cCMP monomers; for example, they increased by 70% and 27%, respectively, in the absence of amino acids at pH 11 (Fig. S39). This shows how the oligomerisation mechanism via activated 2′,3′-cyclic nucleotides (Fig. 2d) offers the unique ability to recycle hydrolysed RNA-oligomers back into the oligomerisation and ligation chemistry.

In summary, we show that amino acids catalyse the formation of RNA under dry alkaline conditions at ambient temperature. Our experiments suggest amino acids enhance oligomerisation by acid-base catalysis without requiring additional chemical activators, thereby enhancing prebiotic plausibility. The catalytic effect, especially for hydrophobic amino acids, produces a more balanced nucleobase composition in oligomeric RNA creating diverse sequences, eventually enabling the hybridisation of RNA into double strands.

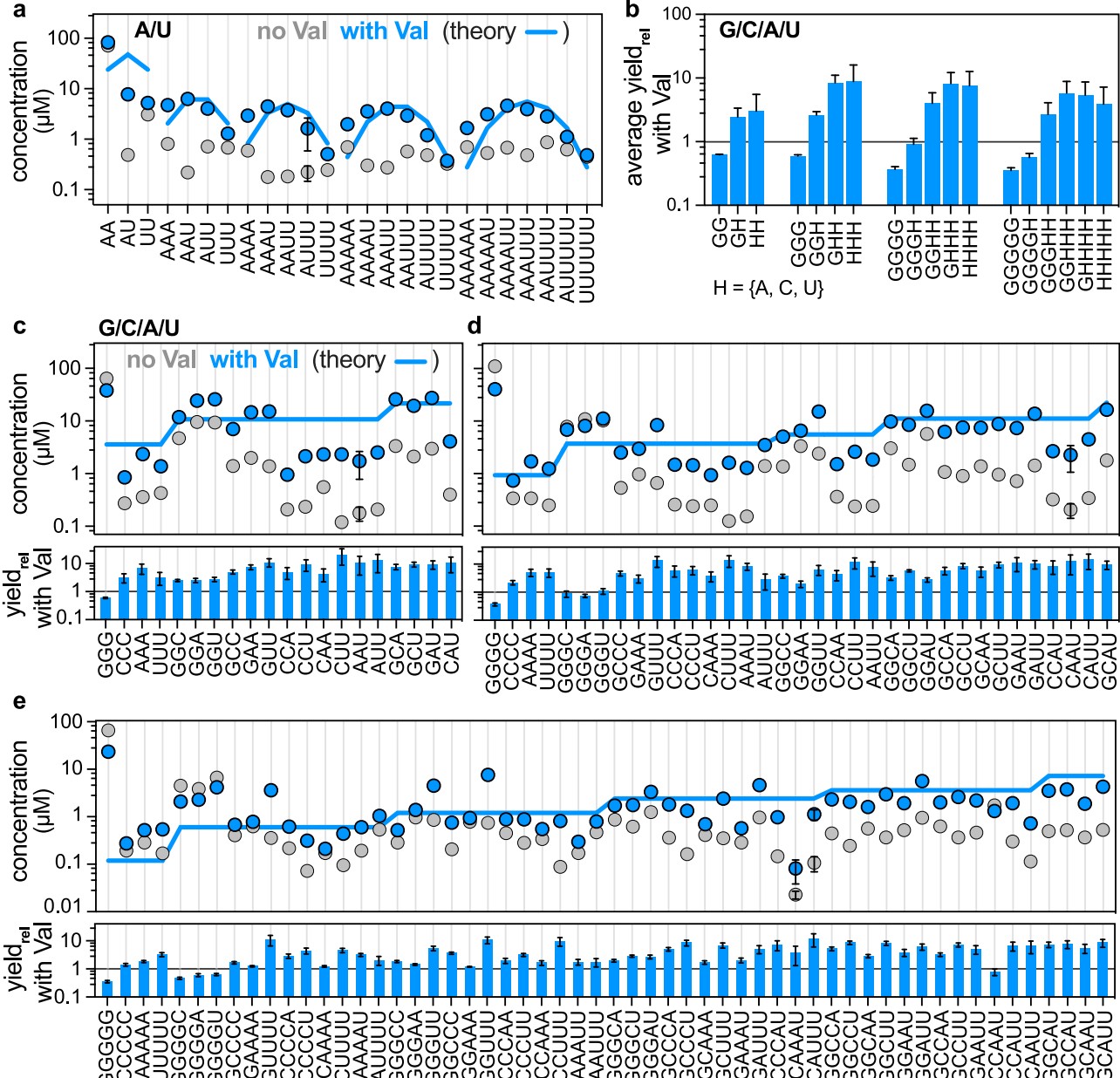

**Fig. 4 | Enhanced compositional diversity in the amino acid-catalysed co-oligomerisation of A/U and G/C/A/U nucleoside 2′,3′-cyclic phosphates.**
**a** Concentrations of the A/U-oligomers of different lengths and compositions are plotted in the absence (grey circles) and presence (blue circles) of valine on a log scale. The blue trace represents random oligomerisation in which A and U are equally efficient. **b** Average relative yield (ratio of the RNA yield in the presence and absence of valine) is presented for 2–5 mer RNA with differential G incorporation in G/C/A/U copolymerisation. The average is calculated considering all compositions in the set (for example: HH includes AA, AU, AC, CC, CU and UU). **c–e** The RNA oligomers (3–5 mer) are expanded for various compositions, in which the concentrations (top) of the oligomers in the absence (grey) and presence (blue) of valine are presented. The blue trace represents random oligomerisation, where all

nucleotides are assumed to have equal efficiency. The distribution ratio of the oligomers is adjusted based on the total yield of the n-mers in the performed reaction. The bottom plot shows the relative yield (fold increase) for each sequence composition. In the experiments, 40 mM cyclic nucleotides and 100 mM valine at pH 10 and 50 μl were rapidly dried and incubated for 20 h at 25 °C. The quantifications were done by LC-MS and a self-written LabVIEW programme. The results are the mean with S.D. of three independent experiments, and we show the largest S.D. in each data set (complete errors in Figs. S25a and S31). The isotopic distribution AAAAA and GGGCC is identical, and both concentrations are assumed equal, marked by * in (**e**). All isotope fits are given in Supplementary Data 3. Source data are available for all panels (**a–e**).

The absence of divalent ions such as $Mg^{2+}$ can help resolve the RNA strand separation problem during templated replicative cycles. Furthermore, the RNA oligomers with 2′,3′-cyclic phosphate termini have been shown to undergo high-fidelity copying by templated ligation under the same pH conditions[33]. The experiments required low salt freshwater at pH 9, 10 (i.e., the $pK_{aH}$ of amino acids), which are, for example, abundantly found in low-temperature volcanic environments

of Iceland[46]. Interestingly, geological phosphate enrichment, which is required for cNMP synthesis, also correlates with alkalinity in carbonate-rich lakes[47]. These observations are consistent with volcanic islands on the early Earth[48], providing (periodically) drying conditions that would be conducive to the formation and copying of RNA. The porous rocks found in such environments would provide protection against UV and offer air-water interfaces[41], and a protected wet-dry

cycling by day-night oscillations that has been successfully probed for this RNA polymerisation chemistry[49,50]. The air-water interfaces could drive strand separation for continuous replication by templated ligation with ribozymes[51,52] and would enable fast sequence evolution toward increasing length[53]. Moreover, it was recently demonstrated that geothermal heat flows drive the separation of precursor biomolecules[54]. Interestingly, amino acids such as Val, Leu, and Ile, which enhanced RNA polymerisation the most, were also selectively enriched by heat fluxes. The functional link found between RNA and amino acids provides a first step to uncover the cross-catalytic mechanisms between peptides and RNA for early precursors of translation and the evolution of the genetic code[55–57].

## Methods

A comprehensive, fully referenced description of the materials and methods is provided in the Supplementary Information. Below is a concise summary of the experimental and analytical procedures employed in the study.

### Dry-state polymerisation of ribonucleoside 2′,3′-cyclic phosphates

RNA polymerisation reactions were performed under dry-state conditions using commercially available ribonucleoside 2′,3′-cyclic phosphates and amino acids. Reaction mixtures were prepared using nucleotide and amino acid stock solutions in nuclease-free water, followed by pH adjustment with KOH or HCl. Small volumes of the mixture were dried on glass slides or in multi-well plates under airflow and incubated at room temperature for 20 h. RNA products were extracted with water and, when necessary, further purified by ethanol precipitation using ammonium acetate and glycogen as co-precipitants.

### HPLC ESI-TOF mass spectrometry

RNA oligomers were analysed using high-performance liquid chromatography (Agilent 1260 Infinity II) coupled with electrospray ionisation time-of-flight mass spectrometry (Agilent 6230B with Dual AJS ESI). Oligomers of varying lengths were separated by reverse-phase ion-pairing chromatography using an Agilent AdvanceBio Oligonucleotide C18 column (4.6 × 150 mm, 2.7 μm) maintained at 60 °C, with a gradient elution at a flow rate of 1 mL/min. The mobile phases consisted of water (Bottle A) and a 50:50 methanol-water mixture (Bottle B), each containing 8 mM triethylamine (TEA) and 200 mM hexafluoroisopropanol (HFIP). Calibration for product concentration was based on pre-formed standards of RNA oligomers. Mass spectrometry data were analysed using a custom LabVIEW-based programme[58] (see Fig. S40 in the Supplementary Information).

### NMR measurements

One-dimensional $^{31}$P NMR spectra were acquired on a Bruker Avance III 300 MHz spectrometer. The data were analysed using Bruker TopSpin 4.3.0 to determine the regioisomeric composition of the RNA linkages (3′−5′ and 2′−5′).

### Computational modelling of amino acid-catalysed RNA polymerisation

We employed three complementary computational approaches to investigate amino acid-catalysed RNA polymerisation: classical quantum chemical calculations, classical molecular dynamics (MD), and ab initio MD simulations (detailed in the Supplementary Information). The datasets are available at Zenodo[59].

Geometry optimisations and activation energy calculations were carried out at ωB97XD/def2TZVPD level of theory on simplified transphosphorylation models involving ethanol as the nucleophile, and β-1-dimethylamino-ribose-2′,3′-cyclic phosphate as a nucleotide analogue. Bulk solvation effects were described with the polarisable

continuum model (PCM) using cyclohexane ($\varepsilon = 2.0$) and DMSO ($\varepsilon = 46.8$) to mimic hydrophobic and hydrophilic (low-water) environments, respectively. All quantum chemical calculations were performed with the Gaussian09 computer code.

Classical molecular dynamics simulations were conducted using the AMBER22 programme combined with the OL3 and ff14SB force fields for nucleotides and amino acids, respectively. These simulations explored amino acid distribution around the cCMP molecules to rationalise the experimentally observed selectivity in RNA polymerisation. Hydrogen bonding and interaction patterns were analysed using cpptraj and VMD.

Ab initio molecular dynamics simulations were performed with CP2K on minimal hydrated systems containing cCMP, amino acids, and a limited number of water molecules to assess amino acid-selective interaction patterns relevant to catalysis in dry-state RNA polymerisation.

## Data availability

All relevant data supporting the findings of this study are provided within the article and the supplementary information. The Supplementary Information contains a comprehensive materials and methods section, Figs. S1–S40, and Table S1, offering further data on morphological characterisation (optical microscopy), quantitative molecular analysis (HPLC-ESI-TOF, NMR), and theoretical modelling. A source data file is provided, which includes data in the main figures in spreadsheet form. The Supplementary Data feature the mass spectra of all products along with isotope distribution fits. The raw MS datasets produced and examined in this study are available from the corresponding author upon request due to the size restrictions. Datasets from the molecular dynamics simulations and quantum mechanical calculations can be accessed at https://doi.org/10.5281/zenodo.15307808. Source data are provided with this paper.

## Code availability

The LabVIEW programme code (Spectral_browser_3.58) used for Mass Spectrometry analysis in this study is available at: https://doi.org/10.5282/ubm/data.588.

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

## Acknowledgements
This work was supported by the European Research Council (ERC) Evotrap, grant no. 787356 (D.B. and S.W.); the Deutsche Forschungsgemeinschaft (DFG, German Research Foundation)—Project-ID 364653263—TRR 235 (CRC 235) (D.B., S.W. and C.B.M.) and Project-ID 521256690—TRR 392 (CRC 392) (D.B., S.W., C.B.M. and S.K.R.), the Simons Foundation (grant no. 327125) (D.B.), the Excellence Cluster ORIGINS funded by Germany's Excellence Strategy EXC-2094-390783311 (D.B., C.B.M. and S.K.R.), support from the Volkswagen Foundation (D.B., C.B.M. and M.W.P.), the Czech Science Foundation (GAČR grant. no. 22-25057S) (J.E.S.) and the Swiss National Science Foundation (grant no. 206871) (S.K.R.). We thank Roland Riek and Jason Greenwald for their help with the NMR measurements and Jiří Šponer for providing the computational platform for the modelling.

## Author contributions
Project design, funding acquisition and supervision: S.K.R. and D.B.; experimental design and data acquisition: S.K.R., S.W., M.W.P. and J.E.S.; computational modelling: M.K., G.C. and J.E.S.; data analysis: S.K.R., S.W. and D.B.; analysis and writing of the manuscript: S.K.R., S.W., J.E.S., C.B.M., M.W.P. and D.B.

## Funding

## Competing interests
The authors declare no competing interests.
