## [Peer Review file · Nature Communications]

Amino acids catalyse RNA formation under ambient alkaline conditions

Corresponding Author: Professor Dieter Braun

Version 0:

Reviewer comments:

Reviewer #1

(Remarks to the Author)

This work describes the potential link between the informational polymer RNA and amino acids, demonstrating the enhanced RNA oligomerisation and sequence diversity by amino acid in an alkaline dry state on early earth. The data is well presented, and it should be published with minor corrections.

1) The authors found hydrophobic amino acids (Val, Leu and Ile) have the maximum enhancement on RNA oligomerisation. Can the authors rule out the pure hydrophobic effect of these amino acids on the system during the drying process? The competing reaction of oligomerisation is hydrolysis due to considerable amount of water in the system. Did the authors find any mono-nucleotides-2'-phosphate or 3'-phosphate in the mixtures after reactions? If yes, were there more of those in the reaction with the hydrophilic amino acids compared to hydrophobic amino acids? It will be more convincing if the authors provide the comparisons of microscopic structures of dry state for the reactions with hydrophobic amino acids or hydrophilic amino acids, to see if the hydrophobic side chains lower the activities of water.

2) It will be easier for the reader if the authors can clarify 1 / 2 is the deprotonated form of 1* / 2* in Figure 2.

Reviewer #2

(Remarks to the Author)

In this study, Braun and co-workers ask if amino acids can catalyze RNA formation in a non-templated fashion. They explore a variety of conditions and determine that alkaline conditions are essential for amino acid-assisted catalysis. Impressively RNA polymerization from 2',3' cyclic phosphate monomers is increased 100-fold under certain amino acid and alkaline conditions. Also impressive is that, in the presence of valine, the rate of incorporation of all four bases is similar, leading to polymers of near equal base composition, enhancing the diversity of the resultant RNA. The faster polymerizing bases, like G, were slowed down by some amino acids, while the slower polymerizing bases, like A and C, were sped up. Such diversity in composition of the resultant RNA is important for the evolution of diverse catalyst candidates. This is an interesting paper but I have a number of comments.

Major Comments:

1. The authors state that the mechanism for oligomerization is general acid-base catalysis with a protonated amine serving as the general acid (Line 300). However, the amine is a lousy acid, with a pKa of 9.6, meaning that even when it forms, it is reluctant to give up its proton.
2. The authors discuss alkaline conditions favoring "high fidelity copying by templated ligation" in ref 32 (line 256 and elsewhere). However, high pH leads to deprotonation of G and U on the WCF face. This would inhibit base pairing with the template. How can this be resolved?
3. This paper relies on the premise that the amino group of the amino acids can act as a catalyst (lines 297-300), but amines are very reactive and likely reacted with the very abundant carboxylates on early Earth. Is there evidence that amines would persist on early Earth as opposed to being tied up as amides?
4. Histidine is perhaps the most likely amino acid to carry out general acid-base catalysis, given its pKa near 7. But it doesn't (Figure 1). Why doesn't it catalyze the reaction the best? The authors should present any data from calculations, or speculate.

Additional Comments:

1. RNA in general is less stable at high pH. How do the alkaline conditions, which drive RNA formation, affect the stability of the RNA that they are trying to make? There seems to be data in Fig S26 that yield is improved at high pH when valine is present, but I don't know if there is any data about suppression of hydrolysis?
2. Abstract. The authors state, "The enhancement of oligomerisation was nucleobase-selective, resulting in increased compositional diversity necessary for subsequent molecular evolution." I was confused when I read this because it seemed self-contradictory. Upon finishing the paper, I realize that the authors mean "The enhancement of oligomerization was most for the least reactive nucleobases causing a leveling of polymerization, resulting in...". I suggest the authors reword like this or something similar to avoid confusing/losing the reader.
3. Line 54. The "RNA world" does not mean that amino acids were not important. This is a false argument / logical fallacy. The "RNA world" simply means that at some point in time in the evolution of life RNA played the key role in storing and replicating information; it doesn't mean that RNA didn't have any help from amino acids or other metabolites. In my opinion, the outcomes of the authors' study support the RNA world. The paper doesn't need to lay claim to this tired technicality of "RNA world = RNA-only world" to be impactful.
4. In several places in the manuscript (Abstract, line 277) the authors say that length dependent concentrations are enhanced up to 100-fold by amino acids, but it could be much more than this for the 6mer and 7mers. One of the challenges in catalysis is measuring the rate of very slow background reactions (see following reference). Can the authors comment on whether they tried measuring the background rate for no amino acid 6 and 7mers. Radzicka, A. & Wolfenden, R. (1995). A proficient enzyme. *Science* 267, 90-93.
5. Figure 1c. What is being plotted as yield here? Is it of one certain oligomer length? Or is it summed over all oligomers, and if so, is each oligomer multiplied by its length?
6. Line 117. I couldn't follow this sentence. Do the authors mean that 5-7mer of C are ~20% of the yield?
7. Line 118. Phe appears to contribute 10-fold to polymerization of C, Fig 1C, in opposition to this statement of no enhancement (of any base).
8. Line 121. It isn't informative to state that when G polymerization with amino acids goes down it is "likely due to interference with its self-catalysis by amino acids". Is there any other possibility? More helpful would be to discuss how arginine, which inhibits G polymerization the most, can pair with the Hoogsteen face of G, or in the previous comment that Phe can stack with the bases and interfere with some kind of organization/alignment.
9. Line 123. Why might the more hydrophobic amino acids give greater product yields? Might they avoid the RNA the most and so not inhibit the reaction? Since the authors are writing a single Results and Discussion section, I expected some analysis to be made "on the fly" here.
10. It isn't clear to me how Figures 1 and S4 differ. The data look identical.
11. How were the assignments into 3'-5' and 2'-5' made in Figure S7. If standards were used, do they work for chimeric oligos (i.e. with a mixture of the linkages in a single oligo).
12. Figs S8-11. It isn't clear how the longer oligomer assignments were made. For example, in Fig S8, there is a "C3" label, but no peak is apparent.
13. Fig 2. It is unclear to me why, in the presence of lower concentrations of valine (10 and 25 mM), pH 12 gives better yield than pH 9 (panel C). Also, it isn't clear why at 50 mM valine, pH 12 has a much poorer rate than pH 9 (panel B) but the same yield (panel C).
14. Line 195. When the authors write, "The simulations show that the amino acids exhibiting higher catalytic activity in experiments (Val, Leu, Ile, Lys) have a noticeably higher preference to interact with the phosphate group than those found to be less active." (a) I think they mean "the amino group of the amino acids exhibiting...". Is this true? (Figure 2E suggests this). If so, it should be stated as such. (b) Could "preference to interact with the phosphate group" be due to avoidance of interaction with other parts of the RNA (see Comment 9 above)?
15. Lines 220-225. Given the surprise of similar concentration of 6mers and 3mers, and the notion that G and U might catalyze copolymerization, is there any bias towards certain A3U3, e.g. AUAUUAU over AAAUUU or UUUAAA?
16. Line 243. This line makes it sound as if 2'-5' linkages hydrolyze faster than 3'-5' linkages. Is that true?
17. Fig S33-S37. Does the pH 10 control have 20 h of drying too? I didn't think so, but then why would its trace look different +/- valine?
18. Line 246. Does this refer to pH 11 data?
19. Lines 248-249. the "ability to recycle hydrolyzed RNA-oligomers" could allow larger RNAs to grow rapidly. Can the authors give us a scenario here? e.g. two tetramers react with each other to make an 8mer, and this is enhanced because hydrolyzed oligos can be recycled to make even longer RNAs.
20. The authors should provide the temperature of the reaction into each Figure legend. They start their conclusion on line 251 about being at ambient temperature but little was said on this in the Results.
21. There is lots of data at 4 oC but almost nothing was said about it.

Minor Comments:

1. Reference 13 was superseded by the following reference, which I encourage the authors to explore. Gong, B., Chen, J. H., Chase, E., Chadalavada, D. M., Yajima, R., Golden, B. L., Bevilacqua, P. C. & Carey, P. R. (2007). Direct measurement of a pK(a) near neutrality for the catalytic cytosine in the genomic HDV ribozyme using Raman crystallography. *J Am Chem Soc* 129, 13335-13342.
2. There are large groupings of references, especially in the Introduction. Two comments: (a) Is it necessary to have so many (e.g. 1-8, 20-32, and so on). And (b) is the order in which these references appear meaningful? If they are placed in a random order, I encourage the authors to put the most important first or place them in chronological order.
3. I'm not sure the great level of detail on the 1973 Orgel paper, including buffer conditions, is necessary, especially in the Introduction. I was convinced the present study was original without it. I suggest this paragraph either be removed or moved to an appropriate section of the Results or Discussion.
4. The supporting figures are not called out in order. e.g. Fig S6 was not called.

(Remarks to the Author)

The manuscript entitled: "Amino acids catalyse RNA formation under ambient alkaline conditions" shows a synergistic effect of amino acids and the cyclic-2',3'-nucleotides for oligomerization process. From the prebiotic perspective, this is a great idea to show how small molecules (plausibly prebiotic) can evolve and help one of the most difficult processes to achieve (oligomerization).

Overall, the manuscript describes, step by step, a full protocol for the cyclic nucleotide's oligomerization. The methodology is well described as the instruments and the software (protocol for quantification) that were used.

Major concerns arise about this study:

1. It is well known that the 2',3'-cNMP are plausibly prebiotic components of the primordial soup, however, its hydrolysis is well documented as well. In this case after 20 h (at higher pH), the competition between hydrolysis vs oligomerization is not well defined in the manuscript. Only they claimed: "It should be noted that the amino acids appear to induce minimal hydrolysis of 2',3'-cyclic phosphates during drying (Fig. S18), suggesting an amino acid-dependent hydrolysis pathway in the drying process", but the plot on figure S18 is just about Val. Figure S7 shows the ³¹P NMR reaction spectrum of the four 2',3'-cNMP with valine at pH 10, but is just the region from -1.5 ppm to 1.0 ppm. It would be good to show the rest of the spectrum, some questions arise: what happened with the signal of the cyclic nucleotide? What about the signals of the free 2'- or 3'-NMP (hydrolysis products)? Another good thing will be to show the ³¹P NMR {H-coupled} version. Finally, for quantification, what kind of standard did they use? Did they run quantitative ³¹P-NMR?

2. Overall, valine (Val) gave the best yields, the improvement for C, A, U and G was according to the manuscript 122x, 49x, 6.8x and 1.5x respectively. About the others amino acids (aa), the improvement was not high or even a little detrimental (in the case of 2',3'-cGMP). The "role" that the aa are playing are no clear. Based on the figure 2d and e, the amino group of the aa is the important one for the so call "transition state", where this NH₂ group made a H-bond with oxygen atoms of the phosphate group. All the aa in theory can form the H-bond, but Val gave the best results. Based on the *paH values Ala (9.87), Gly (9.78), Ile (9.76), Leu (9.74) and Val (9.74) have similar values, but it is no clear the reason that Val performed well. No mention if maybe the hindrance or the bulkiness of the R group of the aa could have a role as well. This explanation could be more extended in order to understand the aa role. The data of the pKaH was taken from: (<https://www.vanderbilt.edu/AnS/Chemistry/Rizzo/stuff/AA/AminoAcids.html>).*

3. *The title of the manuscript is quite ambiguous, because only Val showed improvement if we compare the yields when the reaction carried out without the aa, even though (figure 2c) shows that for oligoG, except for Val, Leu and Ile, the others aa were detrimental for the oligomerization yields. The manuscript mentioned that: "The maximum enhancement was observed with aliphatic hydrophobic amino acids (Val, Leu and Ile), and we detected A- and C-homo-oligomers up to seven nucleotides long, whereas in the absence of these amino acids, only trace amounts of tri- or tetramers were detected" but after mentioned that: "Like valine, glycine and lysine most effectively enhance the oligomerisation yields around pH 9-10, albeit to a lesser extent". However, that improvement is not clear, no reported any yield. There are some inconsistencies with this claim:*

a. Based on figure 2c (conditions pH 10, after 20 h), for oligoC, OligoA and oligoU Leu and Ile performed a little higher than Val (never more than 10%).

b. In the case of oligoG Val, Leu and Ile performed similarly. According to this plot Gly and Lys did not improve the oligomerization yields.

Based on the points mentioned above the title and the conclusion need to be revised and changed to fit more with the results.

4. *Using amino acids as a catalyst for cyclic nucleotide oligomerization is a good idea, but there are some considerations for this study.*

a. To know the intrinsic activity of the aa as catalyst the authors should mention in the manuscript Turnover Frequency (TOF) or at least mentioned in a more detailed manner the role of the aa.

b. There is a study (Chem Systems Chem 2023, 5, e202200026) where oligomerisation of mixed Nucleotide 2',3'-cyclic monophosphate was carried out from a 20 mM solution at 40 °C for 18 hours. The study claims: Our data suggests that cGMP oligomerises in dry state at moderate temperatures and pH. The oligomerisation occurs over a range of temperatures (40–80 °C) and pH (7–12) and does not require additional catalysts, making this reaction robust. Figure 2a showed a plot like figure 1b of this manuscript. In both cases oligoG polymerize until nt 10, and the rest nucleotides follow the same pattern cGMP>cUMP>cAMP>cCMP. The conditions are completely different (from 40 °C in the previous study to room temperature in this new study). It seems that the yields improved from about 0.35% to 39% in the case of oligoG. If this is the case they need to mention because it is a huge change.

c. Another study (Chem Bio Chem 2022, 23, e20220042) used foam induced polymerization of 2',3'-cGMP and 2',3'-cCMP. The wet-dry cycling conditions at the moving interfaces led to the oligomerization of RNA. In this case, the authors used cGMP and cCMP. Figure 6 shows a plot of cGMP oligomerization, oligo length (nt) = 8 and the concentration (μM) like the manuscript under revision.

How does it compare to the established literature? Except for the temperature and the use of amino acids; the way that the data is presented, the quantification technique, and the pH intervals are like the previous studies published before by the same authors. It is my opinion that this manuscript resembles an extension work from earlier research than a novel and innovative study.

Minor concerns or suggestions:

1. Figure S6 was not mentioned in the manuscript. Its title: Estimation of oligomers from the outer and inner area of the dried nucleotide and valine mixture. Based on this plot, only in the oligoG was a substantial difference when the sample for the LC-MS analysis was taken from the inner or outer area. It would be a good idea to correlate this information with figure S1 (this figure shows the morphological characterisation of the dried mixture of cCMP and aa.

2. Taking advantage of the crystallization process that was done during the study, using these crystals to obtain a MALDI spectra for establishing without a doubt the masses and the oligos that were observed.

3. Finally, the conclusion claims: "Amino acids catalyse the formation of RNA under dry alkaline conditions at ambient temperature. Our experiments suggest amino acids enhance oligomerisation by acid-base catalysis without requiring additional chemical activators, thereby enhancing prebiotic plausibility". Only 3 amino acids improved the yields (if compared to reactions without aa) for oligoG and oligoU, the rest aa seem to be a little detrimental for the oligomerization yields (judging the data presented in figure 1b). While for oligoC and oligoA the improvement is only about 7.3 and 8.4%, respectively.

Version 1:

Reviewer comments:

Reviewer #1

(Remarks to the Author)

This work describes the potential link between the informational polymer RNA and amino acids, demonstrating the enhanced RNA oligomerization and sequence diversity by amino acid in an alkaline dry state on early earth. The data is well presented and supporting the conclusions the authors have made. The revised manuscript has clearly addressed the previous concerns we had, and it may be published as it is if the other reviewers have no further concerns.

Reviewer #2

(Remarks to the Author)

Major Comments:

1.) It is probably the case that the reaction goes well at high pH because the population of the general base has gone up and with a pKa of 9.6, this is a very good general base. This is worth considering/mentioning.

Regarding the statement, "as shown in the panels of Figure 2, the yields in our reactions show a strong correlation with the pKaH values of the amino acids, with higher yields observed at $pH > pKaH$." I simply do not see any correlation, let alone a strong correlation, of yields and pKa of the amino acids. What I do see is yield at a maximal yield of ~7% at $pH \sim 10$ for Val (pKa of 9.6), Lys (pKas of 9.0 and 10.5) and Gly (pKa of 9.6). Nearly all these pKa's are the same and the yields are about the same at ~7%. This is not a correlation, which requires a linear relationship of log yield on pKa, whose slope can give the beta value for the reaction. These data don't even provide a thresholding effect. To the extent that the conclusions of the study depend on a correlation, it is on the authors to demonstrate correlation.

2.) If "high fidelity copying by templated ligation" works with the authors previous paper but not with this one, then this needs to be said. As I pointed out in my first review, it doesn't make sense that there would be high fidelity copying, which the authors are saying is because "this study (is) on single-stranded RNA polymerization". Yet the authors write "Furthermore, the RNA-oligomer generated with 2',3'-cyclic phosphate termini are activated towards high fidelity copying by templated ligation under the same pH condition.³³" I didn't understand that this sentence doesn't apply to the present study; rather I thought that it applied to the present study and 33 was support for it. Something should change, such as adding "although it does not apply to the current study" or the reader will be completely misled.

4.) I appreciate the authors doing this additional experiment, which new insight into the reaction. Namely, it shows that histidine likely uses its terminal amine and not its side chain in the reaction. I suggest that this data be added to the manuscript and not just shown as a rebuttal.

Additional Comments

1.) Suppression of hydrolysis does not have to come about by lowering the pH, as the authors' response implies. "As seen in Figure S19, Valine rather has no protective role since with valine the pH remains at the initial pH10, not dropping to lower pH that would protect RNA from hydrolysis." For instance, Valine might interact with the RNA in some way to tie it up. It will be surprising to most RNA biologists to hear that an 8mer has a half-life of 200h at pH 12, even at 5 oC. It would be helpful to include the calculations on the extrapolations from the Breaker paper as SI. The Breaker results had only a single 2'OH and the 8mer has 7, while an 80 mer has 79 and so on. The likelihood of cleaving the RNA at least once scales with the length of the RNA. The implications of nicking a larger RNA and the temperature effects are real (and interesting) and help the reader understand the strengths and limitations of the mechanism.

4.) I understand the difficulty of measuring slow background reactions. Still I would like to challenge the authors to at least consider a linear extrapolation on amino acid concentration back to zero amino acid. They might find more than 100-fold rate acceleration. At the very least they can consider if the "enhanced up to 100-fold" is really "enhanced by at least 100-fold".

Reviewer #3

(Remarks to the Author)

The manuscript entitled: "Amino acids catalyse RNA formation under ambient alkaline conditions" shows a synergistic effect of amino acids and the cyclic-2',3'-nucleotides for oligomerization process. After reading the

updated version of the manuscript, the authors took into account the reviewers' comments and appropriately updated the manuscript.

In my opinion the manuscript is ready to be published, just a final comment. The authors claimed: "However, the reduced hydrolysis at this temperature led to preserving more 2',3'-cyclic phosphates" (Line 242). But based on figure S7, the signals of the hydrolysed cyclic mononucleotides are observed around 4 ppm as 2'- and 3'- monophosphates. These signals are more abundant than the 3'-5' and 2'-5' phosphodiester-linked products and as well the cNMP. I would like to describe in more detail this NMR spectra.

In the same figure, (maybe it is a mistake) in the section c of the figure, OligoG it is visible some temperature of 60 C and some signals as a background (light green), this is part of the figure? Because if that is the case, why only with cGMP and valine did run these experiments?

Finally, around the 20 ppm region (same figure) some small signals are visible (in all cNMP, more visible in the cUMP case), can the authors mention something about these signals?

Version 2:

Reviewer comments:

Reviewer #2

(Remarks to the Author)

The authors have addressed all of my comments in a clear and scholarly fashion, which I appreciate. I recommend publication.

Signed Review: Phil Bevilacqua

Reviewer #3

(Remarks to the Author)

The manuscript entitled: "Amino acids catalyse RNA formation under ambient alkaline conditions" shows a synergistic effect of amino acids and the cyclic-2',3'-nucleotides for the oligomerization process. After reading the updated version of the manuscript, the authors took into account the reviewers' comments and appropriately updated the manuscript. In my opinion the manuscript is ready to be published.

We appreciate the time and effort of all the Reviewers in assessing our manuscript and providing their feedback. Our responses to the raised concerns are highlighted in blue, and sections which show changed parts in the manuscript are in blue and bold.

Reviewer #1 (Remarks to the Author):

This work describes the potential link between the informational polymer RNA and amino acids, demonstrating the enhanced RNA oligomerisation and sequence diversity by amino acid in an alkaline dry state on early earth. The data is well presented, and it should be published with minor corrections.

1) The authors found hydrophobic amino acids (Val, Leu and Ile) have the maximum enhancement on RNA oligomerisation. Can the authors rule out the pure hydrophobic effect of these amino acids on the system during the drying process?

As indicated by the experiments, we believe that the observed enhancement is a combination of the chemistry of base catalysis and the physics of the hydrophobic effect and not purely a hydrophobic effect of the amino acids. This is, for example, indicated by the pH dependence of the reaction (Figure 2a). The amino end is the chemical driving force, with hydrophobicity perhaps aiding in the positioning and potentially reducing the number of water molecules around the 2',3'-cyclic phosphates.

We now write in lines 193-195: **“While the amino ends of the amino acids drive acid-base catalysis, the hydrophobic side chains may contribute to the spatial arrangement and potentially reduce the hydration around the 2',3'-cyclic phosphates.”**

We tried to perform experiments to quantify the remaining water in the dry state, but the used method of thermogravimetric analysis suffers from ambiguities due to the degradation of some of the amino acids. We discuss this below.

The competing reaction of oligomerisation is hydrolysis due to the considerable amount of water in the system. Did the authors find any mono-nucleotides-2'-phosphate or 3'-phosphate in the mixtures after reactions? If yes, were there more of those in the reaction with the hydrophilic amino acids compared to hydrophobic amino acids?

Yes, we did observe the hydrolysis product of the cyclic nucleotides in the reaction mixtures. This part of the analysis is now shown in **Figure S4b** which presents the fraction of cyclic mononucleotides after the reaction. As can be seen, non-hydrolysed cyclic phosphates are reduced in concentration if the reaction uses hydrophobic amino acids:

Also, as can be seen in the S18, at the beginning of the drying phase with Valine (Fig. S18a) and **Lysine (now added as Fig. S18b)**, the starting amount of reactive monomers is reduced due to this hydrolysis, which can reduce the yield of polymerization. To some extent, this could also explain the reduction in pH after drying (Fig. S19).

We have now added this as a discussion in the main manuscript in lines 177-184:

“It should be noted that the polymerisation efficiency rises without amino acids at a high pH of 11 to 12, likely due to 5'-OH activation (Fig. 2a). However, this is ultimately limited by nucleotide and RNA degradation. At lower amino acid concentrations, the pH dependence persists (Fig. 2c). At higher concentrations, amino acids induce hydrolysis of 2',3'-cyclic phosphates during drying, suggesting an amino acid-dependent hydrolysis pathway already in the drying process (Fig. S18), reducing the overall yield. We normalized against these degradation effects in the analysis of Fig. 2b since the kinetics is measured only after drying. Therefore, we interpret the peak at pH 9 to be the direct result of acid-base catalysis.”

It will be more convincing if the authors provide the comparisons of microscopic structures of dry state for the reactions with hydrophobic amino acids or hydrophilic amino acids, to see if the hydrophobic side chains lower the activities of water.

We tried to perform thermogravimetric measurements to quantify the amount of water in the dry state. However, due to the degradation of some of the amino acids, no direct assessment of water content for all amino acids could be performed. The low amount of samples did not allow for other measurement methods. In the analysis where degradation was minimal—for Leu and Gly— we did not find that the hydrophobic Leu created a dry state with less water content as compared to Gly which showed a lower water content. Thus we have no evidence that hydrophobicity does correlate with lower amounts of water, but as Lys and Val could not be analyzed, the data basis is limited. However, with the above, we tend to believe that it is rather a local effect of positioning the amino acid and keeping away water in the closest vicinity of the catalytic site that is not necessarily related to the overall water content of the dry material.

It will be easier for the reader if the authors can clarify 1 / 2 is the deprotonated form of 1* / 2* in Figure 2.

This has been clarified in the legend of figure 2.

Reviewer #2 (Remarks to the Author):

In this study, Braun and co-workers ask if amino acids can catalyze RNA formation in a non-templated fashion. They explore a variety of conditions and determine that alkaline conditions are essential for amino acid-assisted catalysis. Impressively RNA polymerization from 2',3' cyclic phosphate monomers is increased 100-fold under certain amino acid and alkaline conditions. Also impressive is that, in the presence of valine, the rate of incorporation of all four bases is similar, leading to polymers of near equal base composition, enhancing the diversity of the resultant RNA. The faster polymerizing bases, like G, were slowed down by some amino acids, while the slower polymerizing bases, like A and C, were sped up. Such diversity in composition of the resultant RNA is important for the evolution of diverse catalyst candidates. This is an interesting paper but I have a number of comments.

Major Comments:

1. The authors state that the mechanism for oligomerization is general acid-base catalysis with a protonated amine serving as the general acid (Line 300). However, the amine is a lousy acid, with a pK_a of 9.6, meaning that even when it forms, it is reluctant to give up its proton.

At $pH = pK_{aH}$, half of the amino groups of the amino acids are expected to be deprotonated. However, as the Reviewer commented, protonated amines can act as weak acids and may not be highly effective in donating protons. It should be noted that the reactions were performed in a dry state with limited availability of water. Thus, the effective dry-state pH could be slightly altered in this environment, and the molecular arrangements might be more crucial in facilitating the chemistry. Still, as shown in the panels of Figure 2, the yields in our reactions show a strong correlation with the pK_{aH} values of the amino acids, with higher yields observed at $pH > pK_{aH}$.

2. The authors discuss alkaline conditions favoring “high fidelity copying by templated ligation” in ref 33 (line 256 and elsewhere). However, high pH leads to deprotonation of G and U on the WCF face. This would inhibit base pairing with the template. How can this be resolved?

This concern relates to findings from a previous paper (Serrão, A. C. *et al.*, *JACS*, 2024) in which we demonstrated that the terminal base in a sequence plays a crucial role in determining the outcome of RNA ligation. We showed there that the reactions are indeed templated, as evidenced by the strong differences in results with and without a template, also with base pairs including G and U. We also argue there that the pK_a of a base at the templated ligation site and in an RNA duplex does not reflect the pK_a of an isolated base.

This situation is however distinct from the focus of the current manuscript, which investigates the polymerization of single-stranded RNA. Based on our unpublished screening results, we have not found any evidence for templated polymerization thus far. Therefore, we think that these considerations on templated ligation are directly relevant for this study on single-stranded RNA polymerization. The reference should instead demonstrate that, under prebiotic conditions, polymerization and ligation could potentially coexist in a one-pot system, enabling both processes to contribute to RNA formation.

3. This paper relies on the premise that the amino group of the amino acids can act as a catalyst (lines 297-300), but amines are very reactive and likely reacted with the very abundant carboxylates on early Earth. Is there evidence that amines would persist on early Earth as opposed to being tied up as amides?

In prebiotic environments, amino acids and amines were likely abundant, as demonstrated in meteorite studies (Cronin, J.R. & Moore, C.B., *Science*, 1971; Aponte, J.C. *et al.*, *GCA*, 2014) and experiments like the Miller-Urey synthesis (Miller, S.L., *Science*, 1953; Parker, E.T. *et al.*, *PNAS*, 2008).

Even though amines could react with carboxylates, they may not necessarily be tied up as amides on early Earth. This is because the conditions that would drive peptide bond formation (activation of amino acids or catalysis) were likely not present in sufficient quantities. In fact, peptides were relatively scarce. Therefore, the presence of amines as catalysts in early-life chemistry is plausible, even if they could form amides under certain conditions.

4. Histidine is perhaps the most likely amino acid to carry out general acid-base catalysis, given its pKa near 7. But it doesn't (Figure 1). Why doesn't it catalyze the reaction the best? The authors should present any data from calculations, or speculate.

True, but the experiment in Figure 1 was fixed at a pH of 10, far from the pKa of the sidegroup of Histidine. Nevertheless, at this pH of 10, Histidine still enhanced the polymerization and we argue this is the case due to the terminal amine. However, to test, we have now run the polymerisation experiment with His starting at pH 7 (see above) and found no oligomerisation for C/A/U and only a 0.2% yield for G. We think this low (no) yield result at pH 7 is due to the pKa of the 5' OH group, which is around 13, is too high (and distant from 7) to effectively facilitate polymerization under these neutral conditions.

Additional Comments:

1. RNA in general is less stable at high pH. How do the alkaline conditions, which drive RNA formation, affect the stability of the RNA that they are trying to make? There seems to be data in Figure S26 that yield is improved at high pH when valine is present, but I don't know if there is any data about suppression of hydrolysis?

If we understand the question correctly, the Referee refers to Figure S36 (now S37), not S26? The Figure S26 refers to polymerization with and without valine at pH 10. As seen in Figure S19, Valine rather has no protective role since with valine the pH remains at the initial pH10, not dropping to lower pH that would protect RNA from hydrolysis. But we think this is rather important for the protection of the reactive cyclic phosphate monomers. The experiments in Figures S34–S37 demonstrate that RNA cleavage under our low-salt conditions becomes significant only at higher pH levels of 13. For instance, as shown in Figure S37, RNA degradation of formed GC oligomers becomes significant at temperatures >5°C and at pH 13. However, we are far from such extremes under our typical experimental conditions.

At such high pH, the RNA cleavage rate is expected to increase by a factor of 10 per unit of pH (Li, Y. and Breaker, R.R., JACS, 1999). Our results of RNA not degrading significantly are actually in accordance with the estimates provided in the Breaker study, suggesting that hydrolysis is not a dominant factor under the conditions used in our experiments. For example, extrapolating from the Breaker paper, we expect at pH 12 and 5°C, a halftime of an 8mer of around 200h in liquid water. Also it should be noted that the dry-state reaction conditions may further mitigate the RNA hydrolysis.

2. Abstract. The authors state, “The enhancement of oligomerisation was nucleobase-selective, resulting in increased compositional diversity necessary for subsequent molecular evolution.” I was confused when I read this because it seemed self-contradictory. Upon finishing the paper, I realize that the authors mean “The enhancement of oligomerization was most for the least reactive nucleobases causing a leveling of polymerization, resulting in...”. I suggest the authors reword like this or something similar to avoid confusing/losing the reader.

We have now rephrased the sentence in the abstract to: **“The fold-change in oligomerisation yield was nucleobase-selective, resulting in increased compositional diversity necessary for subsequent molecular evolution”**

3. Line 54. The “RNA world” does not mean that amino acids were not important. This is a false argument / logical fallacy. The “RNA world” simply means that at some point in time in the evolution of life RNA played the key role in storing and replicating information; it doesn’t mean that RNA didn’t have any help from amino acids or other metabolites. In my opinion, the outcomes of the authors’ study support the RNA world. The paper doesn’t need to lay claim to this tired technicality of “RNA world = RNA-only world” to be impactful.

We agree. We however thought that the RNA world is typically understood as a scenario where ribozymes and RNA self-replication are essential for the onset of evolution. We understand the point and have revised the sentence from "vital connection is lost in the 'RNA world' model" to "vital connection is lost in an 'RNA-only world' model."

4. In several places in the manuscript (Abstract, line 277) the authors say that length dependent concentrations are enhanced up to 100-fold by amino acids, but it could be much more than this for the 6mer and 7mers. One of the challenges in catalysis is measuring the rate of very slow background reactions (see following reference). Can the authors comment on whether they tried measuring the background rate for no amino acid 6 and 7mers. Radzicka, A. & Wolfenden, R. (1995). A proficient enzyme. Science 267, 90-93.

Unfortunately, we face the challenge that many non-G polymers are not detectable at the polymerization level of 6-mers or 7-mers using our current methods. Our otherwise sensitive ESI-TOF approach cannot assess rates for these cases without amino acids, making such a study unfeasible. Additionally, without refeeding or performing wet-dry cycles, we do not anticipate that extending reaction times in the studied single dry cycle would lead to significantly increased reactivity. The conformational restrictions imposed by the dry state limit further polymerization. However, we observe longer RNA strands in experiments using wet-dry cycling, which will be discussed in a forthcoming paper with other collaborators.

5. Figure 1c. What is being plotted as yield here? Is it of one certain oligomer length? Or is it summed over all oligomers, and if so, is each oligomer multiplied by its length?

We understand the confusion and have changed the Figure caption to clarify this and write **“The yield of RNA oligomerisation for different amino acids and nucleotides are calculated as the sum of nucleotides in oligomer form, i.e. the concentration of oligomers times their respective length.”**

6. Line 117. I couldn’t follow this sentence. Do the authors mean that 5-7mer of C are ~20% of the yield?

This is indeed a bit cryptic - we meant it relative, that is 20% of the yield went into 5-7 mers. We apologize and thank the Reviewer for raising the concern. We have now rephrased the sentence to:

“oligomers containing 5-7 nucleotides formed during cCMP polymerisation in the presence of hydrophobic amino acids contribute around 20% of the total yield.”

7. Line 118. Phe appears to contribute 10-fold to polymerization of C, Fig 1C, in opposition to this statement of no enhancement (of any base).

Yes, again, we meant this as a relative comparison, but we understand how this could be misunderstood. We wanted to highlight that Phe, despite being hydrophobic, doesn't affect the oligomerization as much as the other hydrophobic amino acids do. We have now rephrased the sentence to: **“although Phe is hydrophobic, its aromatic side chain was not observed to significantly enhance the oligomerisation more than other polar amino acids such as Gly/Lys/His/Asp/Asn, in contrast to Val/Leu/Ile.”**

8. Line 121. It isn't informative to state that when G polymerization with amino acids goes down it is “likely due to interference with its self-catalysis by amino acids”. Is there any other possibility? More helpful would be to discuss how arginine, which inhibits G polymerization the most, can pair with the Hoogsteen face of G, or in the previous comment that Phe can stack with the bases and interfere with some kind of organization/alignment.

We appreciate the Reviewer's feedback. Indeed, this part is a bit speculative. We have however performed quantum molecular dynamics simulations to show that the bulky benzyl side chain of phenylalanine often stacks with the cytosine of the substrate nucleotide and hinders the reorganization of the reaction complex towards the products (Figure S22).

We have rephrased the sentence and added a discussion addressing the suggestion. We now write in lines 127-131: **“Various non-covalent interactions between nucleotides and specific amino acids may significantly interfere with the self-catalysis of the nucleobases by hindering the reactive molecular arrangements, as observed in cGMP polymerisation. These interactions may include stacking of Phe and interactions of Arg with the Hoogsteen face of G.”**

9. Line 123. Why might the more hydrophobic amino acids give greater product yields? Might they avoid the RNA the most and so not inhibit the reaction? Since the authors are writing a single Results and Discussion section, I expected some analysis to be made “on the fly” here.

We have added a short discussion of our preliminary thermogravimetric measurements addressing the overall water content of the samples (see our answers to Referee 1) above. However, the results were not conclusive enough. Therefore, we now add the sentence: **“However, hydrophobic amino acids could reduce the overall water content in the dry phase while also interacting with the nucleobases to enhance the positioning of the amino acid at the cyclic phosphate.”**

10. It isn't clear to me how Figures 1 and S4 differ. The data look identical.

Indeed, Figure 1 and Figure S4 show the same data, however in Figure S4 the error bars for each point are shown, which we omitted in Figure 1 for clarity. We have now added a note to the caption of Figure S4 - **“We replot the data of Figure 1b with error bars.”**

11. How were the assignments into 3'-5' and 2'-5' made in Figure S7. If standards were used, do they work for chimeric oligos (i.e. with a mixture of the linkages in a single oligo).

Sorry, this was our mistake. We unfortunately omitted this information in the caption of Figure S7. The peak assignments were performed following the NMR details and peak assignments reported in Motsch et al., 2019. This reference was not cited in the SI, but is now added (Ref. 3 in SI). Also, a new panel (a) has been added to Figure S7 which shows the whole spectra (also asked for by referee 3). In the legend of Figure S7, we now write: **“The peak assignments were performed following SI Ref 3. The signals between -1.5 – -0.5 ppm indicate predominantly dinucleotide products with phosphodiester bonds. Hydrolysed cyclic mononucleotides are observed around 4 ppm as 2'- and 3'- monophosphates. A significant amount of unreacted cNMP is detected near 20 ppm.”** and **“Three peaks in this region represent the reaction products, with a 2',3'-cyclic phosphate end and hydrolysed 2'- or 3'-linear phosphate ends.”**

12. Figs S8-11. It isn't clear how the longer oligomer assignments were made. For example, in Fig S8, there is a “C3” label, but no peak is apparent.

We do not see the peaks since we plot the data here on a linear y-axis scale to allow the stacking of the data with pH. We acknowledge the concern and have removed some labels from the provided Figures S8-11. However, the peak assignment is based on oligomer standards and there the C3 peak in Figure S8 is indeed visible when the data is plotted logarithmically. This is the usual method for us to analyse mass spectrometry data, as seen in the screenshot of the analysis (Materials and methods, MS data analysis, Figure M4). The full peak assignments are documented in full detail in supplementary data 2, where images of the peaks are shown.

13. Fig 2. It is unclear to me why, in the presence of lower concentrations of valine (10 and 25mM), pH 12 gives better yield than pH 9 (panel C). Also, it isn't clear why at 50 mM valine, pH 12 has a much poorer rate than pH 9 (panel B) but the same yield (panel C).

This is indeed a bit of a complex discussion and we have improved it with changed text as shown and discussed below. First of all, the panel (b) is a kinetic evaluation where we subtracted the effect of cyclic deactivation documented in Figures S18 by renormalizing the concentrations right after drying. Panel (c) in contrast is a 20h experiment as usual in the manuscript, where no normalization against the pH and amino acid-dependent degradation of the initial cyclic nucleotide material is performed. Second, the buffering of the pH as we move from the initial pH state to the dry state is a function of the amino acid concentration. Enhanced amino acid concentration buffers the reduction of pH from the degradation of 2',3'-cyclic form to linear 2' or 3' phosphate, well seen in Figure S19. Third, as we increase the pH, we come closer to the base activation of the 5'-OH group, which becomes significant.

With this, we can understand the behaviour of panel (c). At low valine concentration, the increase of yield is only seen for higher pH as we approach the high pKa of the 5'-OH group. As the valine concentration increases, the yield at initial pH 9 does not increase significantly since its pH in the drying process has dropped likely to pH 7.5 (see Figure S19) and thus the catalytic effect of this low amount of valine cannot yet kick in. In contrast, at high pH, the pH drops to probably pH 10 and an already small amount of valine is catalyzing this reaction. This is however mostly an effect of the difference of the initial pH and the pH right before the dry state. This likely is the explanation for the nonlinear increase of yield with concentration of valine.

With above reasoning, we chose a high concentration of valine in panel (b). While this kinetic analysis does not suffer from the degradation of 2',3'-cyclic nucleotides, the control of pH during the drying process is still rather qualitative than quantitative, as seen from Figure S19 when extrapolated to the dry state. Furthermore, at higher amino acid concentrations, the degradation of the 2',3'-cyclic phosphate activation during drying (see Figure S18) becomes more pronounced, particularly at high

pH. This degradation reduces the availability of activated monomers, ultimately decreasing polymerization efficiency before the kinetics of polymerization is measured.

So we understand the frustration of the referee in interpreting the data, which also reflects our own. However, we do not see much chance to disentangle the above effects completely and make a quantitative relation between panel (b) and panel (c). This is now discussed in lines 177-184 of the manuscript where we write: **“It should be noted that though the polymerisation efficiency rises without amino acids at high pH of 11 to 12, likely due to 5'-OH activation (Fig. 2a). However, this is ultimately limited by nucleotide and RNA degradation. At lower amino acid concentrations, the pH dependence persists (Fig. 2c). At higher concentrations, amino acids induce hydrolysis of 2',3'-cyclic phosphates during drying, suggesting an amino acid-dependent hydrolysis pathway already in the drying process (Fig. S18), reducing the overall yield. But we normalized against these degradation effects in the analysis of Fig. 2b since the kinetics is measured only after drying. Therefore, we interpret the peak at pH 9 to be the direct result of acid-base catalysis.”**

14. Line 195. When the authors write, “The simulations show that the amino acids exhibiting higher catalytic activity in experiments (Val, Leu, Ile, Lys) have a noticeably higher preference to interact with the phosphate group than those found to be less active.” (a) I think they mean “the amino group of the amino acids exhibiting...”. Is this true? (Figure 2E suggests this). If so, it should be stated as such. (b) Could “preference to interact with the phosphate group” be due to avoidance of interaction with other parts of the RNA (see Comment 9 above)?

Yes, this is true. With respect to (a) we now write to clarify: **“We then used classical molecular dynamics to assess the propensity of the amino group of the amino acids to form hydrogen bonds with the 2',3'-cyclic phosphate for acid-base catalysis (Figs. 2d,e and Fig. 3, S21-22 and Table S1).”**. With respect to (b) the simulation indeed includes the interaction with the RNA nucleobase. We changed to **“The simulations show that the amino groups of amino acids exhibiting higher catalytic activity in experiments (Val, Leu, Ile, Lys) have a noticeably higher preference to form hydrogen bonds with the phosphate group as well as with the RNA nucleobase.”**

15. Lines 220-225. Given the surprise of similar concentration of 6mers and 3mers, and the notion that G and U might catalyze copolymerization, is there any bias towards certain A3U3, e.g. AUAUUAU over AAAUUU or UUUAAA?

Unfortunately, we have not yet managed to develop sequencing methods for these very short samples and, therefore, cannot assess the sequence differences raised by the referee. While Illumina kits for short RNA are improving and allowing potential ligation to primers in the future, we are currently unable to address this question further. We hope that the observed effects result from some templating mechanism or an influence of the alkaline pK_a of U, but this remains purely speculative at this stage and requires additional experiments.

16. Line 243. This line makes it sound as if 2'-5' linkages hydrolyze faster than 3'-5' linkages. Is that true?

Yes, this is our observation using custom-synthesized polyC RNA. To fully document this interesting result, we have now incorporated the data in the new Figure S33. A similar result has been shortly mentioned by Järvinen et al. (1991): “In contrast, the 2'-O-ionized 3',5'-dinucleoside monophosphates are cleaved less readily than their 3'-O-ionized 2',5'-counterparts”. This reference is now cited in the main text.

17. Fig S33-S37. Does the pH 10 control have 20 h of drying too? I didn't think so, but then why would its trace look different +/- valine?

We have now marked the control experiments better in the Figure and noted this in the Figure caption. In the experiment, the dried polymerized samples were resuspended in nuclease-free water at pH 10–13 and incubated for hydrolysis at different temperatures. The pH 10 control was the control for the hydrolysis reaction and underwent the 20-hour drying process, but no pH adjustment was made. We have now updated the figure caption by removing the second “pH 10” and replacing it with “control- no hydrolysis” for clarity.

18. Line 246. Does this refer to pH 11 data?

As seen in Figure S39, hydrolysis is seen mainly for higher pH, most notably for pH 13 at elevated temperatures. However, the numbers for the monomer cyclic nucleotide yields are given in Figure S39, and we picked the numbers for the non-valine scenario at pH 11. Importantly, RNA cleavage in Figure S39 was observed to yield more cGMP and cCMP monomers; for example, they increased by 70% and 27%, respectively, in the absence of amino acids at pH 11. We now write: “... **for example, they increased by 70% and 27%, respectively, in the absence of amino acids at pH 11 (Fig. S39)**”

19. Lines 248-249. the “ability to recycle hydrolyzed RNA-oligomers” could allow larger RNAs to grow rapidly. Can the authors give us a scenario here? e.g. two tetramers react with each other to make an 8mer, and this is enhanced because hydrolyzed oligos can be recycled to make even longer RNAs.

These results will be thoroughly addressed in an upcoming paper, where we explore both templated and untemplated ligation in wet-dry cycles, potentially incorporating the feeding of new nucleotides. We anticipate that elongation through reactions such as two tetramers forming an 8-mer is feasible. However, particularly in the presence of amino acids, it is crucial to better control the survival of 2',3'-cyclic phosphates or establish mechanisms for their recycling, as these intermediates are significantly degraded during the drying steps (Figure S18). Preliminary results with lysine show promise, but further investigation is required and will be detailed in a future manuscript.

20. The authors should provide the temperature of the reaction into each Figure legend. They start their conclusion on line 251 about being at ambient temperature but little was said on this in the Results.

We have added the temperature of the experiments, which would all be at room temperature at 25°C, to all the Figure captions of the manuscript.

21. There is lots of data at 4°C but almost nothing was said about it.

We thank the reviewer for pointing this out. We have now added the following (lines 240-243): **“At 4 °C, the copolymerisation of cA/UMP (Figs. S24-25: b) and cG/CMP (Figs. S26-29: b) exhibited similar effects to those observed at ambient temperature, though the yields were relatively lower due to slower reaction kinetics. However, the reduced hydrolysis at this temperature led to preserving more 2',3'-cyclic phosphates.”**

Minor Comments:

1. Reference 13 was superseded by the following reference, which I encourage the authors to explore. Gong, B., Chen, J. H., Chase, E., Chadalavada, D. M., Yajima, R., Golden, B. L., Bevilacqua, P. C. & Carey, P. R. (2007). Direct measurement of a pK(a) near neutrality for the catalytic cytosine in the genomic HDV ribozyme using Raman crystallography. *J Am Chem Soc* 129, 13335-13342.

The reference has been added.

2. There are large groupings of references, especially in the Introduction. Two comments: (a) Is it necessary to have so many (e.g. 1-8, 20-32, and so on). And (b) is the order in which these references appear meaningful? If they are placed in a random order, I encourage the authors to put the most important first or place them in chronological order.

The reference groups include important findings from various labs in the field. These groups are extensive as they encompass several concepts and are necessary to keep the text compact. For example, references 21-33 cover the chemical synthesis of ribonucleotides, non-templated RNA polymerization, templated base-by-base polymerization and ligations.

3. I'm not sure the great level of detail on the 1973 Orgel paper, including buffer conditions, is necessary, especially in the Introduction. I was convinced the present study was original without it. I suggest this paragraph either be removed or moved to an appropriate section of the Results or Discussion.

We appreciate the suggestion; however, we think it is important to clarify the literature since this in previous submissions has created a lot of misunderstandings as the consensus opinion on this paper is not fully in line with what experiments were performed in that 1973 Orgel paper.

4. The supporting figures are not called out in order. e.g. Fig S6 was not called.

True. We have now added the following in Lines 135-136: **"The heterogeneity in the morphology of the dried nucleotide-amino acid mixture did not significantly impact the oligomerisation (Fig. S6)."**

Reviewer #3 (Remarks to the Author):

The manuscript entitled: "Amino acids catalyse RNA formation under ambient alkaline conditions" shows a synergistic effect of amino acids and the cyclic-2',3'-nucleotides for oligomerization process. From the prebiotic perspective, this is a great idea to show how small molecules (plausibly prebiotic) can evolve and help one of the most difficult processes to achieve (oligomerization). Overall, the manuscript describes, step by step, a full protocol for the cyclic nucleotide's oligomerization. The methodology is well described as the instruments and the software (protocol for quantification) that were used.

Major concerns arise about this study:

1. It is well known that the 2',3'-cNMP are plausibly prebiotic components of the primordial soup, however, its hydrolysis is well documented as well. In this case after 20 h (at higher pH), the competition between hydrolysis vs oligomerization is not well defined in the manuscript.

Hydrolysis has many aspects in this reaction. First of all, we have investigated strand cleavage (strictly this consists first of a transesterification, then followed to a less extend by hydrolysis of the 2',3'-cyclic phosphate end) in the oligomers formed after 20 hours at high pH, as detailed in Figures S34–S39. Additionally, we identified hydrolysis products of the cyclic nucleotides in the reaction mixtures, with Figure S4b illustrating that a fraction of cyclic mononucleotides remains active after the reactions. This highlights the correlation between hydrophobic amino acids and both nucleotide polymerization and hydrolysis. Interestingly, at higher amino acid concentrations, the degradation of 2',3'-cyclic phosphates during drying (Fig. S18) becomes more pronounced, especially at high pH. This

degradation reduces the availability of activated monomers, ultimately decreasing polymerization efficiency prior to measuring polymerization kinetics.

To sum up these findings in the discussion, we write now: **“It should be noted that though the polymerisation efficiency rises without amino acids at high pH of 11 to 12, likely due to 5'-OH activation (Fig. 2a). However, this is ultimately limited by nucleotide and RNA degradation. At lower amino acid concentrations, the pH dependence persists (Fig. 2c). At higher concentrations, amino acids induce hydrolysis of 2',3'-cyclic phosphates during drying, suggesting an amino acid-dependent hydrolysis pathway already in the drying process (Fig. S18), reducing the overall yield. But we normalized against these degradation effects in the analysis of Fig. 2b since the kinetics is measured only after drying. Therefore, we interpret the peak at pH 9 to be the direct result of acid-base catalysis.”**

Only they claimed: “It should be noted that the amino acids appear to induce minimal hydrolysis of 2',3'-cyclic phosphates during drying (Fig. S18), suggesting an amino acid-dependent hydrolysis pathway in the drying process”, but the plot on figure S18 is just about Val.

Conducting a comprehensive analysis of all amino acids impact on 2',3'-cyclic phosphate hydrolysis would require substantial effort. However, to check the generality towards the other amino acids, we now include the data also for lysine in **Figure S18b**, which confirms our findings. We would expect a behaviour similar to that of valine (Figure S18a) for other aliphatic amino acids (Leu, Ile) during the drying process due to similar physicochemical properties in aqueous solutions. Furthermore, Figure S4b demonstrates the fraction of cyclic mononucleotides remaining after the reactions, showing that hydrophobic amino acids such as Val, Leu, and Ile promote the hydrolysis of cyclic mononucleotides while simultaneously facilitating their polymerization.

Figure S7 shows the ³¹P NMR reaction spectrum of the four 2',3'-cNMP with valine at pH 10, but is just the region from -1.5 ppm to 1.0 ppm. It would be good to show the rest of the spectrum, some questions arise: what happened with the signal of the cyclic nucleotide? What about the signals of the free 2'- or 3'-NMP (hydrolysis products)? Another good thing will be to show the ³¹P NMR {H-coupled} version.

We have now included additional information in Figure S7, providing further details.

Finally, for quantification, what kind of standard did they use? Did they run quantitative ³¹P-NMR?

For quantification, peaks were assigned by comparison with the details reported in Motsch et al. 2019 (Ref. 3 in SI) and subsequently integrated to calculate the 3'-5' to 2'-5' ratio. We now write in the supplement: **“The peak assignments were performed following SI Ref 3. The signals between -1.5 – -0.5 ppm indicate predominantly dinucleotide products with phosphodiester bonds. Hydrolysed cyclic mononucleotides are observed around 4 ppm as 2'- and 3'- monophosphates. A significant amount of unreacted cNMP is detected near 20 ppm.”** and **“Three peaks in this region represent the reaction products, with a 2',3'-cyclic phosphate end and hydrolysed 2'- or 3'-linear phosphate ends.”**

2. Overall, valine (Val) gave the best yields, the improvement for C, A, U and G was according to the manuscript 122x, 49x, 6.8x and 1.5x respectively. About the other amino acids (aa), the improvement was not high or even a little detrimental (in the case of 2',3'-cGMP). The “role” that the aa are playing is not clear. Based on the figures 2d and e, the amino group of the aa is the important one for the so-called “transition state”, where this NH₂ group made an H-bond with oxygen atoms of the phosphate group. All the aa in theory can form the H-bond, but Val gave the best results. Based on the pK_{aH} values Ala (9.87), Gly (9.78), Ile (9.76), Leu (9.74) and Val (9.74) have similar values, but it is not clear the reason that Val performed well.

We will argue below that this difference between the amino acids is likely due to the hydrophobicity of the side chain. But before, we should clarify that while valine was chosen for its high effect and relatively greater prebiotic plausibility, the other amino acids also demonstrated enhanced catalytic activity, especially for the bases A and C which polymerize inefficiently without amino acids and which do not have an alkaline pKa themselves. This is clearly evident in Figure 1c, where the black line represents the reference yield value without amino acids. For example, Leucine and Isoleucine exhibit better values than Valine, and Lysine shows enhancements of 24x, 22x, 2x, and 0.5x for C, A, U, and G, respectively. We added a text to enhance this discussion: **“Thus, the amino ends of the amino acids drive base catalysis, while the hydrophobic side chains contribute to the spatial arrangement and potentially reduce the hydration around the 2',3'-cyclic phosphates.”**

No mention if maybe the hindrance or the bulkiness of the R group of the aa could have a role as well. This explanation could be more extended in order to understand the aa role. The data of the pKaH was taken from:

(<https://www.vanderbilt.edu/AnS/Chemistry/Rizzo/stuff/AA/AminoAcids.html>).

Based on quantum molecular dynamics simulations, a discussion on the role of side chain bulkiness and steric hindrance is provided on pages 30-31 of the Supplementary Information. We now write in the main text: **“We then used classical molecular dynamics to assess the propensity of the amino group of the amino acids to form hydrogen bonds with the 2',3'-cyclic phosphate for acid-base catalysis (Figs. 2d,e and Fig. 3, S21-22 and Table S1).”** The simulation indeed includes the interaction with the RNA nucleobase. We further write **“The simulations show that the amino acids exhibiting higher catalytic activity in experiments (Val, Leu, Ile, Lys) have a noticeably higher preference of the amino group to form hydrogen bonds with the phosphate group as well as with the RNA nucleobase.”**

3. The title of the manuscript is quite ambiguous, because only Val showed improvement if we compare the yields when the reaction carried out without the aa, even though (figure 2c) shows that for oligoG, except for Val, Leu and Ile, the others aa were detrimental for the oligomerization yields.

We disagree since the polymerization of C and A are enhanced by all amino acids and within error bars also for U. Moreover, the polymerisation of G is still enhanced by amino acids such as Val, Leu and Ile (Figure 1c, most right panel). So we do not understand why the reviewer says that only Val showed an enhancement. Please note that the horizontal line in black is the yield without amino acids and the yields with the mentioned amino acids are presented above this black line. Although the fold enhancement of G is less pronounced than that of the other bases, there is still an increase. This finding is the basis for our title. It should be noted that the same mechanism of acid-base catalysis is already implemented by the alkaline pK_a of the bases U and G, so we do understand why the enhancement of amino acids for these bases is lower since the mechanism is already implemented by the base itself. This differential enhancement of the yield leads then to the near-equal distribution of bases in the mixed copolymerization results, creating diverse sequences, eventually enabling the hybridization of RNA into double strands. (Figure 4).

The manuscript mentioned that: “The maximum enhancement was observed with aliphatic hydrophobic amino acids (Val, Leu and Ile), and we detected A- and C-homo-oligomers up to seven nucleotides long, whereas in the absence of these amino acids, only trace amounts of tri- or tetramers were detected” but after mentioned that: “Like valine, glycine and lysine most effectively enhance the oligomerisation yields around pH 9-10, albeit to a lesser extent”. However, that improvement is not clear, no reported any yield.

This is correct. We changed this text to **“While glycine and lysine most effectively enhance the oligomerisation yields for A, U and C around pH 9-10, we found a reduction in polymerization for G. In the presence of valine, the yields of all cNMP oligomerisation improved substantially and peaked around pH 9-10. However hydrophobic amino acids such as valine, leucine and isoleucine still enhanced the polymerisation for G at pH 10 (Fig.1c).”** We hope this clarifies the issue.

a. Based on figure 2c (conditions pH 10, after 20 h), for oligoC, OligoA and oligoU Leu and Ile performed a little higher than Val (never more than 10%).

This is true, but our point is that a number of amino acids are showing the total yield enhancement mentioned in the title. Actually, the yield enhancements in C, A, U, and G polymerisation for Leucine and Isoleucine are slightly better than Valine (Fig. 1c); however, we selected Valine as a representative hydrophobic amino acid due to its prebiotic plausibility. Regarding the comment “never more than 10%”, we would like to point out that the fold enhancement of yields for C, A and U polymerisation with valine was found to be respectively 122x, 49x, 6.8x. Without this, the RNA polymerisation would be rather negligible.

b. In the case of oligoG Val, Leu and Ile performed similarly. According to this plot Gly and Lys did not improve the oligomerization yields. Based on the points mentioned above the title and the conclusion need to be revised and changed to fit more with the results.

All nucleotides, except cGMP, showed an improvement in yield with amino acids compared to those without, as indicated by the black horizontal line in Figure 1c. Thus, the general trend applies across all four nucleobases for the hydrophobic amino acids and within error bars for all amino acids for the bases C, A, and U. Thus, we think that the title is correctly describing this result. We have, however, updated the abstract to state: **“The fold-change in oligomerisation yield was nucleobase-selective, resulting in increased compositional diversity necessary for subsequent molecular evolution.”** We have changed the conclusion and write now: **“The catalytic effect especially for hydrophobic amino acids produces a more balanced nucleobase composition in oligomeric RNA, creating diverse sequences, eventually enabling the hybridization of RNA into double strands.”**

4. Using amino acids as a catalyst for cyclic nucleotide oligomerization is a good idea, but there are some considerations for this study.

a. To know the intrinsic activity of the aa as catalyst the authors should mention in the manuscript Turnover Frequency (TOF) or at least mentioned in a more detailed manner the role of the aa.

We understand the concern; however, since the reaction is conducted in a dry environment, the amino acids likely have difficulty diffusing between nucleotides at a rate comparable to a fluid situation. The dry state makes it difficult to apply the concept of Turnover Frequency to our reaction conditions. We currently have no precise insights into the molecular architecture of the reacting species in the dry state; however, we have performed theoretical calculations to understand the amino acid-catalysed RNA polymerisation in detail (Fig. 3, Figs. S20-22). We did not detect any change of amino acids by ESI-TOF, such as covalent hybrids between RNA and amino acids. It is also our frustration that the dry state of the reaction inhibits a more classical analysis of amino acids as catalysts. It should be noted that we do see that wet-dry cycles are further enhancing the product yield, likely by the ability of the reactants to be repositioned and by the rehydration and drying cycle.

b. There is a study (Chem Systems Chem 2023, 5, e202200026) where oligomerisation of mixed Nucleotide 2',3'-cyclic monophosphate was carried out from a 20 mM solution at 40 °C for 18 hours. The study claims: “Our data suggests that cGMP oligomerises in dry state at moderate temperatures

and pH. The oligomerisation occurs over a range of temperatures (40–80 °C) and pH (7–12) and does not require additional catalysts, making this reaction robust.” Figure 2a showed a plot like figure 1b of this manuscript. In both cases oligoG polymerize until nt 10, and the rest nucleotides follow the same pattern cGMP>cUMP>cAMP>cCMP. The conditions are completely different (from 40 °C in the previous study to room temperature in this new study). It seems that the yields improved from about 0.35% to 39% in the case of oligoG. If this is the case they need to mention because it is a huge change.

First of all, we only achieved a 3.5% yield for G without amino acids at the higher temperature of 40°C after 18 hours in the above reference study (Figure 1a, pH 10). In contrast, our current experiments achieved a 26% yield for oligoG at 25°C already without amino acids, likely because of the reduced temperature and faster drying protocol used in this manuscript. Previously, we could not detect polymerisation at these colder temperatures which however are prebiotically important since only lower temperatures allow hybridisation of short RNA strands. This clarification has been now added to lines 112–116. It reads now **“It is also worth noting that we reported a 3.5% yield for cGMP polymerization at 40 °C previously⁴¹ but no polymerisation at 25 °C due to a too slow drying process in that study. In our current experiments, the yield increased to 26% without amino acids and to 39% with valine. Most importantly however, a significantly enhanced yield was now observed for the other bases A, U, and C.”**

c. Another study (Chem Bio Chem 2022, 23, e20220042) used foam induced polymerization of 2',3'-cGMP and 2',3'-cCMP. The wet-dry cycling conditions at the moving interfaces led to the oligomerization of RNA. In this case, the authors used cGMP and cCMP. Figure 6 shows a plot of cGMP oligomerization, oligo length (nt) = 8 and the concentration (µM) like the manuscript under revision. How does it compare to the established literature? Except for the temperature and the use of amino acids; the way that the data is presented, the quantification technique, and the pH intervals are like the previous studies published before by the same authors. It is my opinion that this manuscript resembles an extension work from earlier research than a novel and innovative study.

The detection technique remained largely unchanged between the two manuscripts. However, the previous study did not include amino acids and also operated at much higher temperatures, which would inhibit double stranded RNA hybridization in subsequent steps. The dry state produced no mixed GC sequences (ChemBioChem 2022, 23, e202200423 (Fig. 6b)). In contrast, our results in this manuscript with amino acids (Figs. S27, S29) demonstrate a significant increase in the amount of C-bases within the RNA copolymerization, resulting in significantly more diverse RNA compositions. Moreover, the addition of amino acids markedly improved yields and compositional diversity for A/U, G/C and G/C/A/U copolymerization (Fig. 4 and Fig. S27, Fig. S29). These enhancements, particularly regarding the biological relationships between nucleotides/RNA and amino acids/proteins, represent the central message of this manuscript. Therefore, we strongly believe the present work is both novel and innovative.

Minor concerns or suggestions:

1. Figure S6 was not mentioned in the manuscript. Its title: Estimation of oligomers from the outer and inner area of the dried nucleotide and valine mixture. Based on this plot, only in the oligoG was a substantial difference when the sample for the LC-MS analysis was taken from the inner or outer area. It would be a good idea to correlate this information with figure S1 (this figure shows the morphological characterisation of the dried mixture of cCMP and aa.

This is a good point. We have now added this information to the main text in lines 135-136: **“The heterogeneity in the morphology of the dried nucleotide-amino acid mixture did not significantly impact the oligomerization (Fig. S6).”** We think that the hydrophobic nature of G is the reason for this

difference, forming aggregates on the outside of the dried droplet, similar to the valine patterns seen in Figure S1. These aggregates seem to correlate with enhanced polymerization. However, we think the overall effect is not strong enough in terms of yield to be a significant finding in this study.

2. Taking advantage of the crystallization process that was done during the study, using these crystals to obtain a MALDI spectra for establishing without a doubt the masses and the oligos that were observed.

We tried to perform MALDI imaging; however, we found that after spraying the dye matrix for MALDI after the sample formation only showed a very low ionization yield and made analysis difficult. As a result, we struggled to obtain good mass spectra, with ion counts being 10,000 times lower than expected. If we were to add the dye matrix to the polymerisation mixture, the results of course would not be comparable. Based on this finding, we could not use MALDI to evaluate our experiments, especially since the quantification of MALDI signals is difficult due to the calibration of ionization yields. Instead, we opted for the fractionation of samples from the inside versus the outside and then applied the usual HPLC-ESI-TOF as the best methodological compromise to obtain data on the spatial heterogeneity of polymerization. This test indeed did not show a large difference in yield. This situation may however change once we look into serial wet-dry cycles in the future.

3. Finally, the conclusion claims: “Amino acids catalyse the formation of RNA under dry alkaline conditions at ambient temperature. Our experiments suggest amino acids enhance oligomerisation by acid-base catalysis without requiring additional chemical activators, thereby enhancing prebiotic plausibility”. Only 3 amino acids improved the yields (if compared to reactions without aa) for oligoG and oligoU, the rest aa seem to be a little detrimental for the oligomerization yields (judging the data presented in figure 1b). While for oligoC and oligoA the improvement is only about 7.3 and 8.4%, respectively.

As discussed earlier, we have adapted the conclusion to make it more clear that the hydrophobic amino acids are indeed enhancing polymerization for all nucleotides. We now write in the conclusion: **“The catalytic effect especially for hydrophobic amino acids produces a more balanced nucleobase composition in oligomeric RNA, creating diverse sequences, eventually enabling the hybridization of RNA into double strands.”** Indeed, the absolute yield for oligoC and oligoA are 7.3% and 8.4% respectively. However, in terms of improvement from an uncatalyzed reaction, this is 122x and 49x respectively with valine. Therefore, we think the claims made in the conclusion are justified.

Our goal is not solely to achieve the maximal yield, but rather to demonstrate the differential enhancement provided by the amino acids (with a slight reduction for G) to enable the produced RNA as compositionally balanced feeding material for replication by templated ligation in a subsequent step. We think the major finding of our work is to see how two major classes of molecules in (prebiotic) biology can directly enhance one another at the onset of early evolution.

We appreciate the time and effort of all the Reviewers in assessing our revised manuscript and providing their feedback. Our responses to the additional concerns raised are presented below.

Before addressing the reviewers' concerns, we want to mention that we have added a reference (Dagar et al., RNA, 2020)⁴⁹ to the discussion as we were approached by the authors, which reported on the oligomerization of cyclic nucleotides (2', 3' cAMP and 3', 5' cAMP) by dry-wet cycles, however at relatively lower pH of 8 and a higher temperature of 90 °C.

Reviewer #1 (Remarks to the Author):

This work describes the potential link between the informational polymer RNA and amino acids, demonstrating the enhanced RNA oligomerization and sequence diversity by amino acid in an alkaline dry state on early earth. The data is well presented and supporting the conclusions the authors have made. The revised manuscript has clearly addressed the previous concerns we had, and it may be published as it is if the other reviewers have no further concerns.

We would like to thank Reviewer #1 for the thoughtful feedback and supporting comments.

Reviewer #2 (Remarks to the Author):

Major Comments:

1.) It is probably the case that the reaction goes well at high pH because the population of the general base has gone up and with a pKa of 9.6, this is a very good general base. This is worth considering/mentioning.

*We thank the reviewer for the valuable input. We agree that amino acids serve as good general bases (but are lousy general acids). As illustrated in Figures 2a and 2d, general acid-base catalysis is most effective around pK_{aH} . However, its effectiveness reduces at higher pH due to a decreased concentration of the zwitterionic form of the amino acid. Notably, the catalytic mechanism involves both the acid and base forms of the amino acid, with their respective concentrations reaching a maximum at $pH = pK_{aH}$. To this end, we have written in the main manuscript: **"The initial rate of valine-catalysed RNA-oligomerization peaked at pH 9-10, close to the amine pK_{aH} (9.6) of valine. At this pH, the relative ratio of the base (amine) and acid (ammonium) form of amino acid molecules is 1:1 in the reaction mixture."***

Regarding the statement, "as shown in the panels of Figure 2, the yields in our reactions show a strong correlation with the pKaH values of the amino acids, with higher yields observed at $pH > pK_{aH}$." I simply do not see any correlation, let alone a strong correlation, of yields and pKa of the amino acids. What I do see is yield at a maximal yield of ~7% at $pH \sim 10$ for Val (pKa of 9.6), Lys (pKas of 9.0 and 10.5) and Gly (pKa of 9.6). Nearly all these pKa's are the same and the yields are about the same at ~7%. This is not a correlation, which requires a linear relationship of log yield on pKa, whose slope can give the beta value for the reaction. These data don't even provide a thresholding effect. To the extent that the conclusions of the study depend on a correlation, it is on the authors to demonstrate correlation.

*This is true. Very likely, this was a misunderstanding in the last response on our side. When we wrote in the answer, “Still, as shown in the panels of Figure 2, the yields in our reactions show a strong correlation with the pK_{aH} values of the amino acids, with higher yields observed at $pH > pK_{aH}$ ”, we intended to convey that the peak of yield occurs at the pK_a of 9.0-10.5, with the addition of either Valine, Lysine or Glycine. This did not imply that the measurements would be resolved enough to see **differences in the pK_a** for these three amino acids. We wanted to clarify that distinguishing between the pK_a values of Valine, Lysine, or Glycine based on a shift in the reaction yield peak is beyond the resolution of our experiment and outside the scope of this paper. However, we believe this was not the claim we made in the manuscript when we wrote: **“The initial rate of valine-catalysed RNA-oligomerization peaked at pH 9-10, close to the amine pK_{aH} (9.6) of valine. At this pH, the relative ratio of the base (amine) and acid (ammonium) form of amino acid molecules is 1:1 in the reaction mixture. This leads to the maximum reaction rate because it enables a proton abstraction from the 5'-OH of the nucleophile and the concomitant protonation of the O2' or O3' oxygens of the substrate, which serve as the leaving group in the transphosphorylation reaction (Fig. 2d,e). Notably, the resolution of the reaction maxima was not sufficient to correlate with the pK_a of the three amino acids screened here.”***

2.) If “high fidelity copying by templated ligation” works with the authors previous paper but not with this one, then this needs to be said. As I pointed out in my first review, it doesn’t make sense that there would be high fidelity copying, which the authors are saying is because “this study (is) on single-stranded RNA polymerization”. Yet the authors write “Furthermore, the RNA-oligomer generated with 2',3'-cyclic phosphate termini are activated towards high fidelity copying by templated ligation under the same pH condition.³³” I didn’t understand that this sentence doesn’t apply to the present study; rather I thought that it applied to the present study and 33 was support for it. Something should change, such as adding “although it does not apply to the current study” or the reader will be completely misled.

*Yes, we understand the point. We have revised the manuscript from: “Furthermore, the RNA-oligomer generated with 2',3'-cyclic phosphate termini are activated towards high fidelity copying by templated ligation under the same pH condition.³³” to: **“Furthermore, the RNA oligomers with 2',3'-cyclic phosphate termini have been shown to undergo high fidelity copying by templated ligation under the same pH condition.³³”**. We hope that this clarifies the point.*

4.) I appreciate the authors doing this additional experiment, which new insight into the reaction. Namely, it shows that histidine likely uses its terminal amine and not its side chain in the reaction. I suggest that this data be added to the manuscript and not just shown as a rebuttal.

Thank you for your suggestion. We have included this data in the supplementary material (Figure S4c) and noted that it represents a preliminary assessment of conditions. We have not fully screened other amino acids at neutral pH yet. If successful, this would be indeed very interesting, but the case of Histidine actually shows that we do not see signs of optimism there. The likely cause is the significantly distant pK_a of the 5' OH group, which is estimated at around 13.

Editorial note: The equation below is reprinted with permission from Li, Y. & Breaker, R. R. Kinetics of RNA Degradation by Specific Base Catalysis of Transesterification Involving the 2'-Hydroxyl Group. *J. Am. Chem. Soc.* **121**, 5364–5372 (1999). Copyright 1999 American Chemical Society.

Additional Comments

1.) Suppression of hydrolysis does not have to come about by lowering the pH, as the authors' response implies. "As seen in Figure S19, Valine rather has no protective role since with valine the pH remains at the initial pH10, not dropping to lower pH that would protect RNA from hydrolysis." For instance, Valine might interact with the RNA in some way to tie it up.

True. We recognize that this sentence might be indeed too strong in our earlier response, and other mechanisms could be at play. We have not found independent experimental evidence supporting Valine-RNA interactions that would protect the oligomer. Therefore, we agree that we should exercise more caution in this regard.

It will be surprising to most RNA biologists to hear that an 8mer has a half-life of 200h at pH 12, even at 5 °C. It would be helpful to include the calculations on the extrapolations from the Breaker paper as SI. The Breaker results had only a single 2' OH and the 8 mer has 7, while an 80 mer has 79 and so on. The likelihood of cleaving the RNA at least once scales with the length of the RNA. The implications of nicking a larger RNA and the temperature effects are real (and interesting) and help the reader understand the strengths and limitations of the mechanism.

We agree. We have now added the following discussion with the formula to Supplementary Figure S37: "It should be noted that at such a high pH, the RNA cleavage rate is expected to increase by a factor of 10 per unit of pH (see Ref. 40 in the main manuscript, Li, Y. and Breaker, R.R., JACS, 1999). Our results of RNA not degrading significantly at very high pH are actually in accordance with the estimates provided in the Breaker study, suggesting that hydrolysis is not a dominant factor under the conditions used in our experiments. For example, extrapolating from the Breaker paper, we expect at pH 12 and 5°C, a halftime of an 8mer of around 200h in liquid water. This calculation is based on the formula (e) from above Ref. 40:

$$k_{\text{projected}} = k_{\text{background}} \times 10^{\{0.983(\text{pH}-6)\}} \times 10^{\{-0.24(3.16-[\text{K}^+])\}} \times 69.3[\text{Mg}^{2+}]^{0.80} \times 3.57[\text{K}^+]^{-0.419} \times 10^{\{0.07(T_1-23)\}}$$

given in units of 1/min and per base linkage with a background rate of 1.3e-9/min. It can also be rewritten in Excel/Computer syntax with concentrations given in M and Temperature in °C:

$$k_{\text{projected}}=1.30\text{E-}09*10^{(0.983*(\text{pH}-6))}*10^{(-0.24*(3.16-[\text{K}]))}*69.3*([\text{Mg}2+]^{0.8})*3.57*([\text{K}]^{-0.419})*10^{(0.07*(T1-23))}$$

Also, it should be noted that the dry-state reaction conditions may further mitigate the RNA hydrolysis."

4.) I understand the difficulty of measuring slow background reactions. Still I would like to challenge the authors to at least consider a linear extrapolation on amino acid concentration back to zero amino acid. They might find more than 100-fold rate acceleration. At the very least they can consider if the "enhanced up to 100-fold" is really "enhanced by at least 100-fold".

We understand the point; however, our ability to detect the concentration of oligomers, such as 6- or 7-mers, is limited since mass spectrometry analysis is typically restricted to concentrations above 0.1 μ M. In Figure 1, extrapolating the black traces (without amino acids) suggests that the enhancement for 6-mers would be much higher than a factor of 100. For oligoC or oligoA, one could estimate even greater enhancements for a 7-mer (based on the extrapolated black lines in Figure 1b) for C and A in the presence of amino acids. However, we did not want to do too much speculative extrapolation.

We based the factor of 100 on real experimental data, specifically for 3- or 4-mers of A or C in Figure 1b, rather than relying on what might be an overly optimistic extrapolation. That said, we believe that for longer oligomers, the enhancement inferred from extrapolation should be quite higher.

It is important to note that the above does not imply an extrapolation to zero amino acid concentrations. The measurements in Figure S17 primarily reflect shorter oligomers, as the total material plotted is largely dominated by dimers in the oligomer pool. Therefore, we would not be able to measure longer RNA oligomers for direct comparison; doing so would require an exponential extrapolation, which we consider too speculative.

In light of this discussion, we have updated the Figure 1 caption in the manuscript to “at least 100-fold.”

Reviewer #3 (Remarks to the Author):

The manuscript entitled: “Amino acids catalyse RNA formation under ambient alkaline conditions” shows a synergistic effect of amino acids and the cyclic-2',3'-nucleotides for the oligomerization process. After reading the updated version of the manuscript, the authors took into account the reviewers' comments and appropriately updated the manuscript. In my opinion the manuscript is ready to be published, just a final comment.

The authors claimed: "However, the reduced hydrolysis at this temperature led to preserving more 2',3'-cyclic phosphates" (Line 242). But based on figure S7, the signals of the hydrolysed cyclic mononucleotides are observed around 4 ppm as 2'- and 3'- monophosphates. These signals are more abundant than the 3'-5' and 2'-5' phosphodiester-linked products and as well the cNMP. I would like to describe in more detail this NMR spectra.

We thank the reviewer for the comment. We would like to clarify that Line 242, which reads, “However, the reduced hydrolysis at this temperature led to preserving more 2',3'-cyclic phosphates,” refers to the cA/UMP (Fig. S25b) and cG/CMP (Figs. S27b and S29b) co-oligomerization results at 4 °C. The reaction kinetics at this temperature are relatively slower than at ambient temperature, which is also reflected in the lower yield.

In contrast to this, the NMR experiments were conducted using oligomers obtained from reactions at room temperature. Importantly, the NMR analysis focused on quantifying the ratio of 2'-5' vs. 3'-5' linkages in the oligomers, which are predominantly dimers. While this provides insight into the most dominant linkage pattern, the restriction to dimers does not

allow for accurate quantification of polymerization yield. For fully quantitative analysis, we had to rely on mass spectrometry, where samples were frozen immediately after the experiment to minimize backbone cleavage or a prolonged polymerization. The different handling required for NMR samples may have led to a comparably higher hydrolysis of 2',3'-cyclic nucleotides, resulting in a lower recorded yield. Also, the reviewer is correct in noting that the higher peak size of the 2' or 3' linear phosphates around 4 ppm indicates a lower polymerization yield. To address these points, we have now updated Figure S7: **“The significantly higher peaks of the 2' or 3' linear phosphates (around 4 ppm) compared to those representing phosphodiester linkages (-1.5 to -0.5 ppm) suggest a lower oligomerization yield, likely due to the increased hydrolysis of cyclic nucleotides. Unlike quantitative mass spectrometry, which accounts for all possible oligomer lengths in yield calculations, NMR signals for phosphodiester-linked products primarily reflect dimers within the oligomer pool. Additionally, the oligomerization yield of approximately 8% for A, C, and U in the presence of valine (Mass spectrometry data) indicates that a considerable portion of monomers remained unreacted.”**

In the same figure, (maybe it is a mistake) in the section c of the figure, OligoG it is visible some temperature of 60 C and some signals as a background (light green), this is part of the figure? Because if that is the case, why only with cGMP and valine did run these experiments?

We thank the reviewer for raising this issue. It is not a mistake in the plot. To quantify the ratio of phosphodiester linkages, we first acquired NMR spectra at room temperature (Figure S7b). However, we realized that oligoG was not visible in the spectra due to its tendency to form aggregates and thus much broader NMR spectra peaks. To address this, we reacquired the NMR spectra after adding 8M urea to the oligomer samples. The linkage ratio in U, C, and A remained unchanged, though signal quality for A improved slightly (Figure S7c). However, for oligoG, the signals remained inconclusive in determining the linkage ratio (green). To obtain a better spectrum, we tried to solubilize the oligoG sample by measuring it separately at 60°C in the presence of 8M urea (light green).

*We have written in Figure S7: **“oligoG forms aggregates that were not observed by NMR. The NMR signal was improved in 8 M urea and at 60 °C (light green in panel c); however, these spectra were not used to measure the linkage ratio.”***

Finally, around the 20 ppm region (same figure) some small signals are visible (in all cNMP, more visible in the cUMP case), can the authors mention something about these signals?

*We believe this is a dimer RNA with a cyclic phosphate ending (Ref: Thompson, J.E. et al., Biochemistry 1994, 33, 7408-7414). We have now added in S7: **“The signals around 19.5 ppm correspond to dinucleotide products with a cyclic phosphate ending.”***